# Cell fate decision by a morphogen-transcription factor-chromatin modifier axis

Jin Ming[1,2,3,8], Lihui Lin[3,4,8], Jiajun Li [1,2], Linlin Wu[1,2], Shicai Fang[3], Tao Huang[1,2], Yu Fu[1,2], Dong Liu[1,2], Wenhui Zhang[1,2], Chen Li[3], Yongzheng Yang[3], Yi Huang[3], Yue Qin[1,2], Junqi Kuang[1,2,5], Xingnan Huang[1,2], Liman Guo[3], Xiaofei Zhang [3], Jing Liu [3,4], Jiekai Chen [3,4], Chengchen Zhao [1] ✉, Bo Wang [1,6,7] ✉ & Duanqing Pei [1,5] ✉

Cell fate decisions remain poorly understood at the molecular level. Embryogenesis provides a unique opportunity to analyze molecular details associated with cell fate decisions. Works based on model organisms have provided a conceptual framework of genes that specify cell fate control, for example, transcription factors (TFs) controlling processes from pluripotency to immunity[1]. How TFs specify cell fate remains poorly understood. Here we report that SALL4 relies on NuRD (nucleosome-remodeling and deacetylase complex) to interpret BMP4 signal and decide cell fate in a well-controlled in vitro system. While NuRD complex cooperates with SALL4 to convert mouse embryonic fibroblasts or MEFs to pluripotency, BMP4 diverts the same process to an alternative fate, PrE (primitive endoderm). Mechanistically, BMP4 signals the dissociation of SALL4 from NuRD physically to establish a gene regulatory network for PrE. Our results provide a conceptual framework to explore the rich landscapes of cell fate choices intrinsic to development in higher organisms involving morphogen-TF-chromatin modifier pathways.

How a single cell, the fertilized egg, gives rise to all cells in an individual remains an unresolved question[2]. Intuitively, the entire process is quite simple, i.e., a series of cell divisions giving rise to the total number and types of cells in an individual, such as the estimated two hundred trillion cells in a human being[3]. The binary process of doubling has been well understood under the framework of cell cycle[4]. How cell fate decision is specified at each division remains largely unknown at the molecular level. Genes involved in cell fate decisions have been identified in model organisms mostly. For example, the complete cell lineage of *C. elegans* enabled the identification of genes involved in an

extreme case of cell fate decision[5], cell death, which laid the foundation of subsequent biochemical delineation of the cell death pathway[6]. the fruit fly has provided an early link between chromosome aberration and developmental defects such as the homeotic mutations leading to the realization of developmental programming hardwired in the genome, i.e., the co-linearity of HOX clusters and body segmentations conserved in both invertebrates and vertebrates[7–9]. However, unlike cell death which can be modeled quite faithfully in vitro, developmental cell fate decisions to generate diverse cell types have not been modeled successfully in vitro. A fertilized mouse egg

[1]Laboratory of Cell Fate Control, School of Life Sciences, Westlake University, Hangzhou, China. [2]Institute of Biology, Westlake Institute for Advanced Study, Hangzhou, China. [3]CAS Key Laboratory of Regenerative Biology, South China Institute for Stem Cell Biology and Regenerative Medicine, Guangzhou Institutes of Biomedicine and Health, Chinese Academy of Sciences, Guangzhou, China. [4]Guangdong Provincial Key Laboratory of Stem Cell and Regenerative Medicine, Center for Cell Lineage and Development, Guangzhou Institutes of Biomedicine and Health, Chinese Academy of Sciences, Guangzhou, China. [5]Westlake Laboratory of Life Sciences and Biomedicine, Hangzhou 310024 Zhejiang, China. [6]Zhejiang University of Science and Technology, School of Information and Electronic Engineering, Hangzhou, Zhejiang, China. [7]Key Laboratory of Biomedical Intelligent Computing Technology of Zhejiang Province, Hangzhou, Zhejiang, China. [8]These authors contributed equally: Jin Ming, Lihui Lin. ✉e-mail: zhaochengchen@westlake.edu.cn; wangbo@westlake.edu.cn; peiduanqing@westlake.edu.cn

undergoes the first time cell fate decision, giving rise to trophecto-derm or TE and inner cell mass or ICM, around E4.5-E5.5, mouse ICM makes the second fate decision to generate hypoblast or primitive endoderm and epiblast[10,11]. Compared to trophectoderm and epiblast, primitive endoderm is rarely analyzed at molecular details like the epiblast, which ESCs serve as a faithful in vitro copy, but could be as critical as the other cells in embryogenesis. For example, PrE releases the nodal signaling molecule, which affects the anterior-posterior axis specialization process during embryonic gastrulation[12]. Knockout of PrE related genes, for instance, *Sox17* and *Lama1*, arrests embryonic development[13,14]. Recent studies performed on blastoids indicate that PrE over-differentiation might cause intrauterine development lethality[15]. However, the underlying mechanism of PrE lineage specifi-cation remains unknown. BMP signals have been reported to play an important role in PrE based on studies using dominant negative forms of BMP receptor 2 and small molecule antagonists[16]. Similarly, *Sall4* is required for the development of the epiblast and primitive endoderm[17], *Sall4* KO embryos arrest around E6.5 slightly later than the embryo implantation process[18], a stage controlled by PrE. However, the relationship between *Sall4* and *Bmp4* has not been reported so far in any cell fate decision process. Here we report an in vitro model of cell fate decision whereby BMP4 specifies PrE fate by dissolving the pluripotent-bound SALL4-NuRD complex[19]. This system may allow detailed biochemical analysis to delineate the molecular mechanisms associated with cell fate decisions.

## Results

### BMP4 blocks pluripotency induction

We have recently shown that JGES(*Jdp2-Glis1-Esrrb-Sall4*) can convert E13.5 mouse embryonic fibroblasts or MEFs to naive pluripotency (-E3.5 inner cell mass or ICM) in a NuRD-dependent manner[20]. We then wished to improve this process by testing various factors previously known to enhance reprogramming. We surprisingly find that BMP4, a component of the TGF-b superfamily, previously shown to improve OSKM reprogramming[21,22], inhibits JGES reprogramming dramatically (Fig. 1a–c). This unexpected finding suggests that BMP4 might have diverted the cell fate trajectory away from pluripotency along the reprogramming pathway. To gain further insight into this process, we ask if BMP4 inhibition is time-dependent and show that there is a window of sensitivity in the first 3 days (Fig. 1d). We also show that the system is dose-dependent (Fig. 1e, f), with -1 ng/ml capable of inhibit-ing -50%. These results demonstrate that BMP4 blocks JGES repro-gramming to pluripotency in a time and dose-dependent manner.

We then try to address whether BMP4 relies on the known downstream regulators such as the SMADs, a family of proteins known to mediate TGF-b super-family signaling either positively or negatively[23]. Specifically, we show that *Smad6* and *Smad7*, inhibitory or I-Smad antagonize BMP4 effectively and restore reprogramming[24] (Supplementary Fig. 1a, b). It is of interest to note that *Smad6* has a better rescue efficiency than *Smad7* as it is more specific to BMPs[25]. These results suggest that as a classic morphogen, BMP4 specifies the

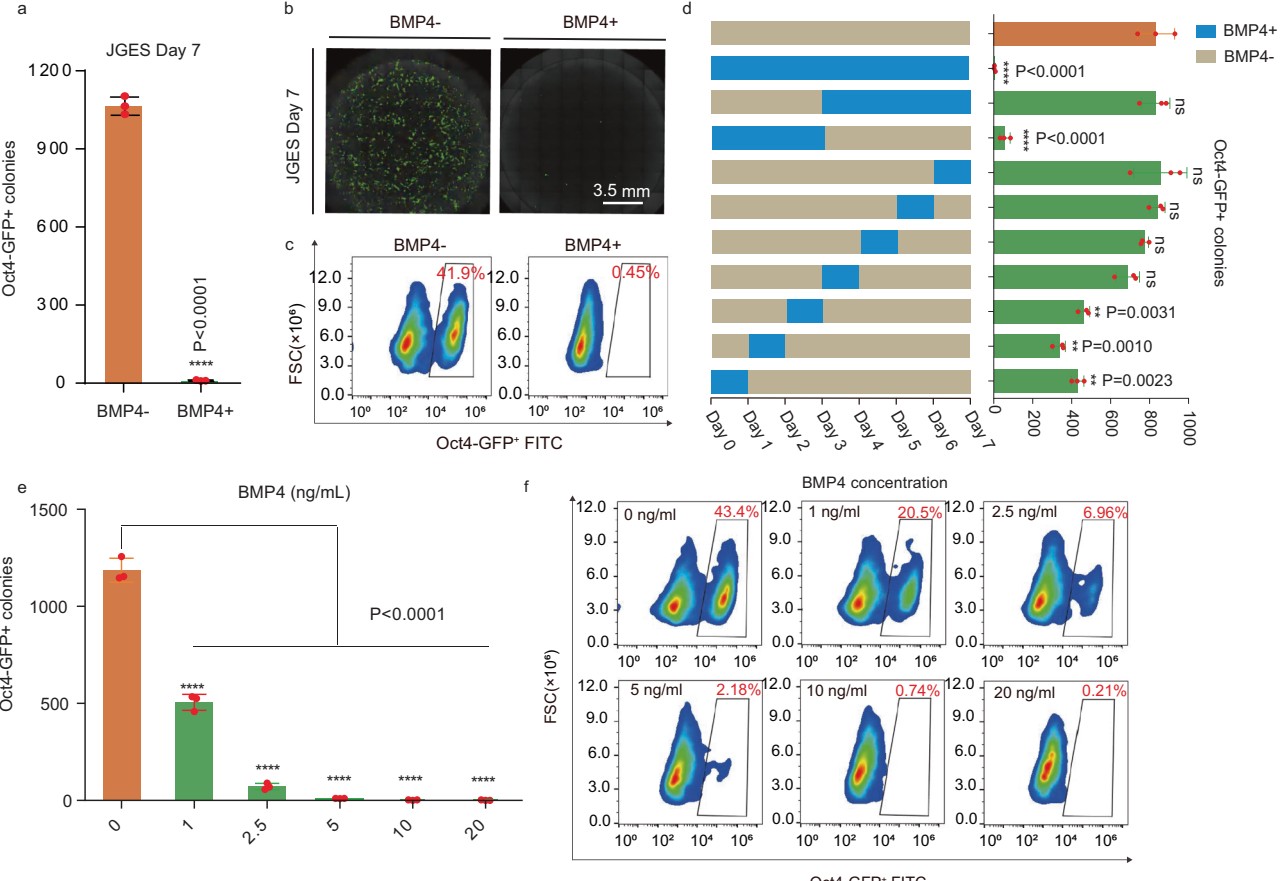

**Fig. 1 | BMP4 blocks pluripotency induction. a** Bar plot for *Oct4* GFP positive iPS colonies numbers of JGES reprogramming under BMP4+ and BMP4− conditions at Day 7, data are mean ± s.d., two-sided, unpaired *t* test; *n* = 3 independent experi-ments, ****p < 0.001, error bars here represent mean with SD. **b** Whole well screening photograph of (**a**), scale bar = 3.5 mm. **c** Flow cytometry results show the *Oct4* GFP positive iPS cell percentage of (**a**). **d** Schema chart and the histogram show *Oct4* GFP positive iPS colonies numbers under BMP4 treatment at different time point, data are mean ± s.d., two-sided, unpaired *t* test; *n* = 3 independent experiments, *p < 0.05, **p < 0.01, ***p < 0.001, error bars here represent mean with SD. **e** Histogram shows *Oct4* GFP positive iPS colonies numbers under BMP4 treatment at different concentrations; data are mean ± s.d., two-sided, unpaired *t* test; *n* = 3 independent experiments, *p < 0.05, **p < 0.01, ***p < 0.001. **f** Flow cytometry results show the *Oct4* GFP positive iPS cell percentage of (**e**).

choice between pluripotent and alternative fates during JGES reprogramming.

## BMP4 diverts cell fate toward extra-embryonic lineages

To identify the fate trajectory diverted by BMP4, we performed single-cell RNA-seq on JGES reprogrammed cells at day 7 with or without BMP4 as illustrated (Fig. 2a) and show that there is significant overlap among various intermediates and endothelial cells between BMP4− and BMP4+ samples (Fig. 2b, c and Supplementary Fig. 2a). However, there is a clear separation of the pluripotent cluster in BMP4− vs. primitive endoderm cell-like cells (PrECLCs) cluster in BMP4+ population (Fig. 2b, c). In addition, there are also minor clusters specific for placenta-like cells in the BMP4+ population (Fig. 2c).

Detailed analysis further reveals four major clusters with representative marker genes labeled on the top (Fig. 2d), including (1) the intermediate cluster is related to ossification and kidney development, (2) the endothelial cluster is related to vasculogenesis and endothelial development, (3) the placenta-like cluster is related to placenta development and epithelial cell morphogenesis, and finally (4) the PrECLCs cluster is related to pattern specification process and endoderm development (Fig. 2e). Interestingly, among the clusters in BMP4+ and BMP4− cells at day 7, ~20% of cells are either PrECLCs or iPSCs respectively (Fig. 2f). It is of interest that, in the absence of BMP4, both PrECLCs and endothelial-like cells can be identified, albeit at much lower frequencies, 0.016% and 0.058% (Fig. 2f), suggesting that the JGES reprogramming is capable of generating quite diverse cell types without BMP4.

To further define the mechanism specifying pluripotency vs. PrE[16,17], we screen factors for PrECLCs based on public internal datasets (Fig. 2g). By qPCR (Supplementary Fig. 2b) and bulk RNA-seq (Supplementary Fig. 2c), we show that genes such as *Sox17* and *Gata4/6* are highly enriched in PrECLCs[26,27]. To see if these genes play any potential role in the bifurcating decision between pluripotent and PrE fates, we tested each gene in JGES and showed that *Gata4* is a critical inhibitor in blocking pluripotent reprogramming (Fig. 2h).

We further validated several critical PrE markers by immunofluorescence and show that GATA4+/LAMA1+ clones are present in JGES reprogramming[28] (Fig. 2i). Indeed, we can identify PrECLCs clones at day 9 in both BMP4− and BMP4+ JGES reprogramming (Supplementary Fig. 2d) with clear boundaries, plump cell morphology, and very condense extracellular matrix, a critical characteristic of primitive endoderm[29]. In an effort to match these in vitro generated PrECLCs with mouse embryonic cells, we compared them to those reported in E4.5–E5.5 embryos (Supplementary Fig. 2e, f) and show that, indeed PrECLCs cluster with primitive endoderm (PrE) and pluripotent cells cluster with epiblast. Additionally, we find that PrECLCs are closer to PrE than parietal endoderm (PaE) or vesical endoderm (VE) in vivo (Supplementary Fig. 2g), confirming them as PrE cell-like cells or PrECLCs. In summary, the JGES reprograming system could reset MEFs into pluripotent states or alternative fates, such as those from an extra-embryonic lineage, in a BMP4-sensitive manner.

## BMP4 targets SALL4 to specify alternative fates

The fact that BMP4 clearly inhibits JGES reprogramming suggests that it mediates cell fate decisions in a TF-dependent manner. Indeed, we show that BMP4 dramatically enhances OS (*Oct4*+*Sox2*) reprogramming efficiency in iCD3 as previously described[30] (Fig. 3a), thus ruling out any role iCD3 may play in the inhibitory effect. We then focused on each individual TF by performing drop-out experiments with *Jdp2*, *Glis1*, *Esrrb* and *Sall4* as illustrated (Fig. 3b), and show that we can rule out *Jdp2* and *Glis1*, but not *Esrrb* and *Sall4* for they are both important in pluripotency induction (Fig. 3c). Since dropping either *Esrrb* or *Sall4* lowers reprogramming efficiency to such a negligible level, we introduce *Oct4*, a factor not involved in

BMP4-mediated inhibition but important in pluripotency induction, to JGES, and show that BMP4 remains capable of blocking JGESO reprogramming by reducing the efficiency by ~70% (Fig. 3d). We repeated the dropout experiments with JGESO and show that *Sall4* is the only factor conferring sensitivity to BMP4, remarkably, dropping out *Sall4* in fact renders the remaining JGEO responsible to BMP4 positively (Fig. 3e). BMP4 also enhances OS + JGE reprogramming as expected (Fig. 3f). Alternatively, we tested each of JGES one by one in OS and show that BMP4 enhances reprogramming, except when *Sall4* is added (Fig. 3g). When *Sall4* is added to OS, BMP4 become an inhibitor of reprogramming, and even when J, G, and E are present alone or together (Fig. 3h). These results demonstrate clearly that BMP4 targets *Sall4* to block iPSC reprogramming.

To clarify BMP4−SALL4 axis in pluripotency inhibition and PrE formation, we perform bulk RNA-seq on JGE, JGES, JGEO, JGESO under BMP4+ and BMP4− at day7, when SALL4 exists, BMP4 exhibits inhibition effect on pluripotent genes and promotion effect on PrE genes (Supplementary Fig. 3a–d). We repeated the same set of dropout experiments with BMP4 treatment again, qPCR results show that *Jdp2* is the only factor to inhibit PrE cell fate, the other three factors, including *Sall4*, work cooperatively to enhance PrE cell fate (Fig. 3i). On the other hand, when adding JGES one by one to OS under BMP4 treatment, *Sall4* turns out to be the only factor that enhances PrE gene expression (Fig. 3j). These results suggest that the BMP4−SALL4 axis acts synergistically to impede pluripotency, and promote PrE fate.

## BMP4 dissociates SALL4 from NuRD

We have previously shown that NuRD is important to orchestrate iPSC reprogramming in JGES, in contrast to its reported role as a barrier in OKSM reprogramming. We further reported that NuRD is recruited by SALL4 to close somatic loci via its N-terminal 12 AA residues[20]. To test if BMP4 may disrupt the NuRD−SALL4 axis, we performed IP-MS on SALL4 in BMP4+ and BMP4− groups on day 3 (Fig. 4a) and show, by volcano map, that SALL4-NuRD interaction is disrupted (Fig. 4b, Supplementary Data 1). Components of NuRD are significantly down-regulated in BMP4+ IP-MS experiments (Fig. 4b) and confirmed in Co-IP experiments (Fig. 4c). These results indicate that BMP4 may block iPSC reprogramming by disrupting the SALL4−NuRD axis.

To directly test if the disrupted cooperation plays a crucial role, we constructed inducible fusion constructs between SALL4 and three components of NuRD, i.e., GATAD2B, MTA1 and MBD3 as illustrated in (Supplementary Fig. 4a). Interestingly, while SALL4-MTA1 or MBD3 has minimal effect, SALL4-GATAD2B can significantly enhance iPSC generation, furthermore, when SALL4-GATAD2B, SALL4-MTA1, SALL4-MBD3 fusion constructs are expressed together at the first three days, the reprogramming efficiency could be enhanced in a synergistically fashion (Fig. 4d), (Supplementary Fig. 4b). These results suggest that covalent fusion between SALL4-NuRD can partially restore iPSC reprogramming in the presence of BMP4.

We also tested the SALL4-NuRD cooperation in an alternative way. We took advantage of our earlier finding that the N terminal 12AAs (N12) of SALL4 plays a critical role in cooperation with NuRD in reprogramming. We tested the effect of BMP4 on N12-JDP2, which was previously shown to be able to rescue mutants such as SALL4[K5A], along with Glis1 and Esrrb. As shown in Fig. 4e, BMP4 could also inhibit the reprogramming efficiency of this system by dissociating N12-JDP2 and GATAD2B interaction (Supplementary Fig. 4c). These results suggest that BMP4 blocks reprogramming by disrupting the N12−NuRD interaction. In fact, we show that BMP4 only improves reprogramming when SALL4 no longer can interact with NuRD comparing reprogramming efficiency between SALL4[WT] with SALL4[delN12] and SALL4[K5A]. (Fig. 4f), While SALL4[delN12] fails to promote iPSC generation (Fig. 4g, h), it can also effectively enhance PrE-related gene expression such as *Gata4* (Fig. 4i).

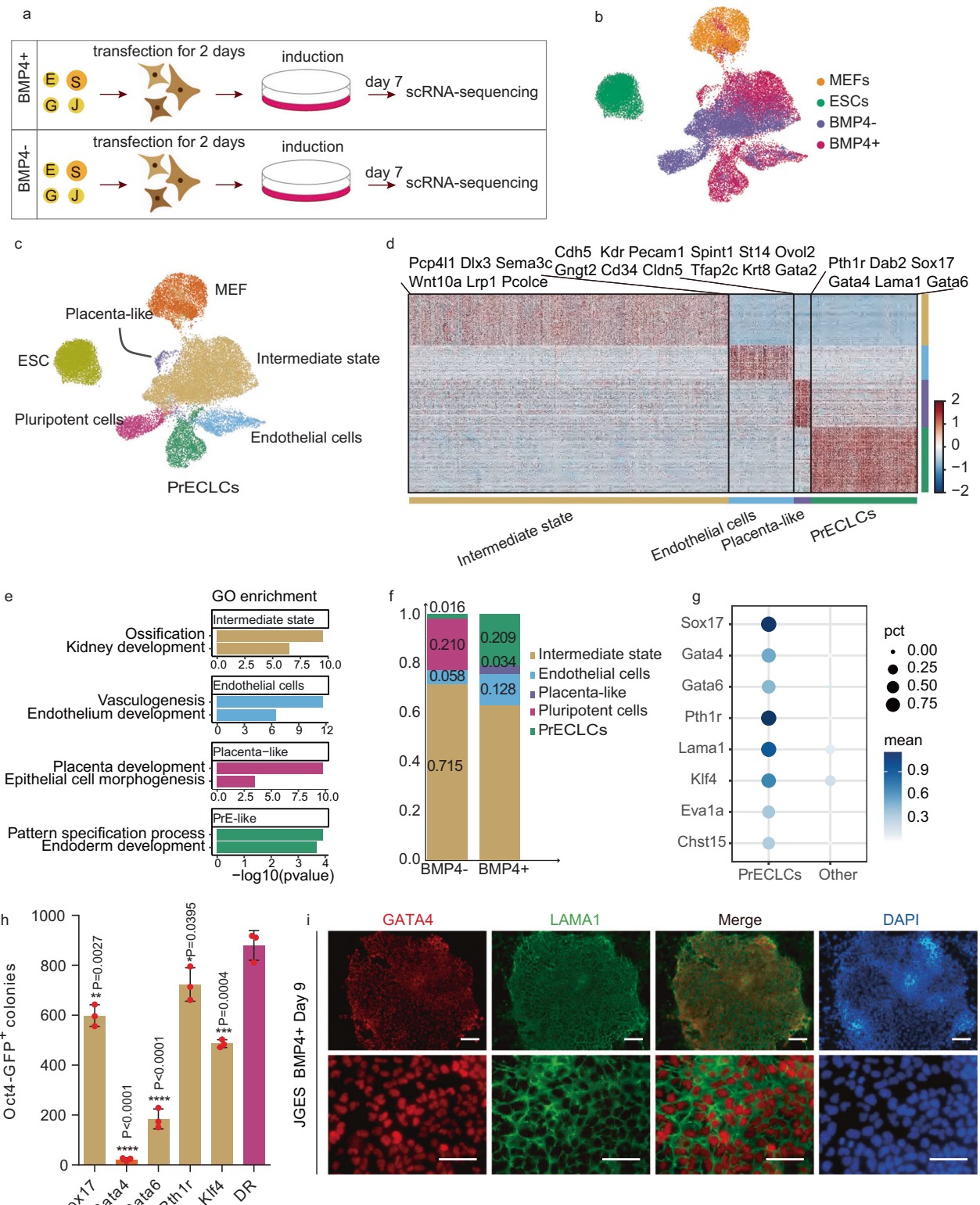

**Fig. 2 | BMP4 diverts cell fate toward extra-embryonic lineages. a** Flow chart shows the single-cell RNA sequencing under BMP4+ and BMP4− conditions after 7 days induction. **b** and **c** UMAP projection of 30,769 cells from MEF, ESC, and JGES reprogramming at day 7, colored by conditions or cell types, MEF and ESC data are from other's work[35]. **d** Heatmap shows differentially expressed genes (DEGs) of four cell types (two-sided *t*-test, *p*-value < 0.01 and log fold change > 0.2). **e** Bar plots showing the representative Gene Ontology (GO) terms. One-sided Fisher's exact test was used to perform enrichment, and terms with Benjamini−Hochberg adjusted *p*-value < 0.1 were selected. The enrichment was performed using the R package clusterProfiler[36]. **f** Stack bar plots showing the proportion of cell types between ±BMP4 conditions. **g** Dot plot showing the mean and percentage (expression = 0, pct) of PrECLCs marker genes between PrECLCs and the other cells at day 7. **h** Bar plot for Oct4 GFP positive iPS colonies numbers of indicated gene expression through JGES reprogramming, data are mean ± s.d., two-sided, unpaired *t*-test; *n* = 3 independent experiments, error bars here represent mean with SD.
**i** Immunofluorescence staining shows the PrECLCs shape clone in BMP4+ condition at day 9 are GATA4 and LAMA1 dual positive, scale bar = 250 μm, *n* = 3 independent experiments.

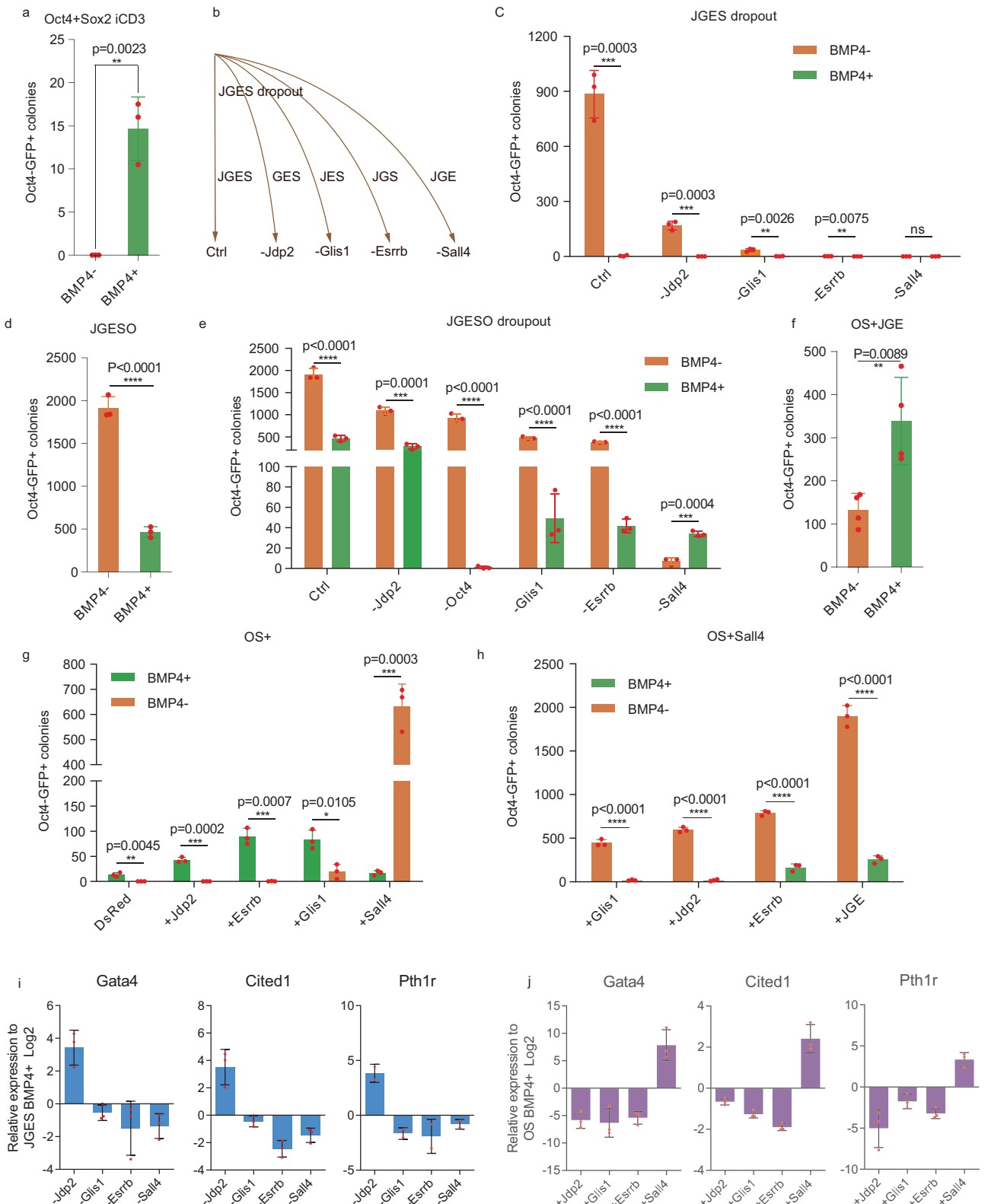

## BMP4 activates PrE regulons

BMP4 is critical for many physiological functions. To probe its role in our system, we performed regulon analysis with our scRNA-sequencing data, and identified five top regulons centered on *Sox17*, *Pitx1*, *Klf4*, *Gata4*, and *Foxa2* in PrECLCs (Fig. 5a). Interestingly, we show that PrE genes are activated by *Gata4* with the rest of the 5 regulons indicated (Fig. 5b–d). Interestingly, *Gata4* is restricted in the PrECLCs cluster. When overexpressed, *Gata4* is the most robust

regulon in blocking the pluripotent fate (Fig. 5e). Consistently, qPCR results show that *Gata4* activates PrE genes significantly more than the other four factors, while *Sox17* and *Foxa2* only elevate PrE gene expression slightly (Fig. 5f). Knocking down *Gata4* leads to PrE genes down-regulation in JGES (Supplementary Fig. 5a, b). Furthermore, we also show that *Gata4* can be elevated by BMP4 (Supplementary Fig. 2b). The mutant SALL4$^{delN12}$, unable to recruit NuRD, can activate PrE genes expression more than SALL4$^{WT}$, thus, phenocopying BMP4

**Fig. 3 | BMP4 targets Sall4 for specifying alternative fate. a** Histogram shows *Oct4* GFP positive iPS colonies numbers of *Oct4* plus *Sox2* reprograming with or without BMP4 treatment, data are mean ± s.d., two-sided, unpaired *t*-test; *n* = 3 independent experiments, *p < 0.05, **p < 0.01, ***p < 0.001. **b** Schema chart shows the single-factor dropout strategy of JGES. **c** Histogram shows *Oct4* GFP positive iPS colonies numbers under different factor dropout in JGES reprograming at day 7 with or without BMP4 treatment, data are mean ± s.d., two-sided, unpaired *t*-test; *n* = 3 independent experiments, *p < 0.05, **p < 0.01, ***p < 0.001. **d** Histogram shows *Oct4* GFP positive iPS colonies numbers of JGES reprograming plus *Oct4* with or without BMP4 treatment, data are mean ± s.d., two-sided, unpaired *t*-test; *n* = 3 independent experiments, *p < 0.05, **p < 0.01, ***p < 0.001. **e** Histogram shows *Oct4* GFP positive iPS colonies numbers under single factor dropout in JGESO reprograming at day 7 with or without BMP4 treatment, data are mean ± s.d., two-sided, unpaired *t*-test; *n* = 3 independent experiments, *p < 0.05, **p < 0.01,

***p < 0.001. **f** Histogram shows *Oct4* GFP positive iPS colonies numbers of OS plus JGE reprograming system with or without BMP4 treatment, data are mean ± s.d., two-sided, unpaired *t*-test; *n* = 3 independent experiments, *p < 0.05, **p < 0.01, ***p < 0.001. **g** Histogram for *Oct4* GFP positive iPS colonies numbers of OS plus DsRed, *Jdp2, Esrrb, Glis1, Sall4* experiments under BMP4+ and BMP4− conditions at Day 7, data are mean ± s.d., two-sided, unpaired *t*-test; *n* = 3 independent experiments, *p < 0.05, **p < 0.01, ***p < 0.001. **h** Histogram for *Oct4* GFP positive iPS colonies numbers of *Glis1, Jdp2, Esrrb*, JGE in OS plus *Sall4* reprograming system under BMP4+ and BMP4− conditions at Day 7, data are mean ± s.d., two-sided, unpaired *t*-test; *n* = 3 independent experiments, *p < 0.05, **p < 0.01, ***p < 0.001. **i** and **j** Histograms show the qPCR results of PrE gene relative expression level of every group, *n* = 3 independent experiments, error bars here represent mean with SD.

(Fig. 4g–i). These results, taken together, suggest that BMP4 activates PrE fate at the expense of pluripotent one through the SALL4−NuRD axis.

### Induction of PrE by SALL4[delN12] alone

The fact that dissociation of SALL4 from NuRD by BMP4 diverts reprograming away from pluripotent to PrE suggests that *Sall4* plays a central role in the fate decision between epiblast- vs. hypoblast-fates. We further hypothesize that *Sall4* alone may be able to specify PrE fate. Previous studies have shown that *Esrrb* has the ability to reset MEF cells to an induced extra-embryonic endoderm (iXEN) state[31]. We also have preliminary evidence that *Sall4, Esrrb* and *Glis1* work cooperatively in PrECLCs induction (Fig. 3i). To test the direct relation between *Sall4* and PrE cell fate, we infected MEFs with single factor SALL4[WT] and SALL4[delN12] and show that GATA4 and LAMA1 double positive PrECLCs clones can emerge from both SALL4[WT] and SALL4[delN12] at day 11[32] (Fig. 6a). To distinguish single factor-induced PrECLCs from JGES induced PrECLCs, we name them iPrEs. However, the earliest iPrE clones appear in SALL4[delN12] at D3 compared to SALL4[WT] at D9 (Fig. 6b). We performed in situ immunofluorescence experiments of GATA4 and show that SALL4[delN12] is indeed more robust in iPrE induction than SALL4[WT] (Fig. 6c, d). These results are consistent with the scRNA-sequence data, in which JGES BMP4− group has a very rare PrECLC population while JGES BMP4+ has a much greater PrECLC population, perhaps as a result of dissociation of SALL4-NuRD by BMP4. We further show by bulk RNA-seq that the PrE markers are expressed in SALL4[delN12] at a higher level than in SALL4[WT] (Fig. 6e, f and Supplementary Fig. 6a).

In order to estimate the characteristics of iPrE, we then construct Sall4[WT]-Flag and Sall4[delN12]-Flag fusion proteins into retroviral vector PMXs, to monitor their expression during iPrE generation and show that both are silenced at D11. Immunofluorescence by FLAG or SALL4 antibodies to detect exo- or whole SALL4 expression suggests that exo-SALL4 is silenced in iPrE colonies and endo-SALL4 slightly activated[33] (Supplementary Fig. 6b). The iPrE cells could be passaged and exhibit robust proliferation in vitro (Supplementary Fig. 6c). Furthermore, the iPrEs generated by SALL4[delN12] proliferate better than those by SALL4[WT]. We also show that in suspended culture, iPrE cells form spherical structures with monolayer cavities indicative of polarity and condensed extracellular matrix, mimicking the PrE property in vivo[34]. Interestingly, these spherical structures can be passaged by trypsin (Supplementary Fig. 6d).

We then performed blastocyst injection experiments to evaluate iPrE developmental potential[27] by marking SALL4[WT] induced iPrE and MEF cells with GFP (Supplementary Fig. 6e) and injecting them into E3.5 blastocysts. The MEFs-GFP cells disappear within 48 h, while iPrEs-GFP remains viable in vitro under identical conditions (Supplementary Fig. 6f). When injected blastocysts shown in (Supplementary Fig. 6g) were implanted into surrogate female mice, we can detect GFP-positive cells in embryo yolk sac in iPrEs-GFP group but none in MEFs-

GFP group at E12.5[11,35] (Supplementary Fig. 6h, i), suggesting that iPrE is capable of integrating into extra-embryonic tissues.

To further characterize SALL4[WT] and SALL4[delN12] iPrE cells, we performed ATAC-sequencing and Cut&Tag experiments and showed that chromatin loci with motifs from FOX, SOX, GATA and KLF family are opened significantly higher in SALL4[delN12] than SALL4[WT] (Fig. 6g, Supplementary Fig. 6i). Results from SALL4 Cut&Tag experiments are quite similar between SALL4[WT] and SALL4[delN12] for PrE genes (Fig. 6h, i), however, H3K27ac is more enriched in SALL4[delN12] than SALL4[WT] among PrE genes, consistent with the fact that the former fails to recruit NuRD (Fig. 6j, k).

## Discussion

We provide here an in vitro model system to analyze early cell fate decisions in development, the choice to becoming epiblast- or hypoblast-cells. Unlike the canonic developmental process of inner cell mass segregating into epiblasts and hypoblasts, we utilize a reprograming approach, converting E13.5 MEFs to pluripotent iPSCs or PrECLCs. We demonstrate that BMP4 plays a crucial role in specifying PrEs away from iPSCs involving SALL4 and NuRD complex.

PrEs are much less understood compared to epiblasts as it lacks an in vitro equivalent as iPSCs or ESCs for epiblasts. But PrEs are emerging as critical as they provide a critical structural components and function of the extra embryonic tissues[36–38]. Regenerating PrE cells through reprograming may provide not only a reliable source of these cells, but also a system to analyze their properties for hypoblast as iPSC or ESC for epiblasts.

Apart from the hypo- and epiblast models, we also uncovered a previously unrecognized cell fate decision axis, linking a classic morphogen BMP4, to a well-known developmentally critical transcription factor SALL4, then to a less well-understood player in development and cell fate control, the NuRD complex. It would be of great interest to see similar paradigms for cell fate decisions in normal development and cancer[39].

Among the axis members, *Sall4* has been shown to play an important role in the three key lineages, epiblast, hypoblast, and trophectoderm, during early embryogenesis[40,41]. Intriguingly, *Sall4* seems to play critical roles in multiple lineages and potentially many cell fate decisions[42], it is conceivable that BMP4 may further provide a signal to switch between various fates and mechanistic actions. This feature may become relevant to assign specific activity of *Sall4* in carcinogenesis as reported previously. As *Sall4* has been reported as a reprograming factor for iPSC in several studies, our results highlight its role in PrE cell fate formation while introducing new inquiries into how SALL4[delN12] or SALL4[WT] remodels MEF cell fate to PrE cell fate. Given the fact that *Sall4* has been shown to take part in limb and genital cell development in ontogenetic process[43–45], along with BMPs or TGF-b family who participate in the same processes[46,47], the cooperative and antagonistic functional research on them are still areas yet to be developed in further studies.

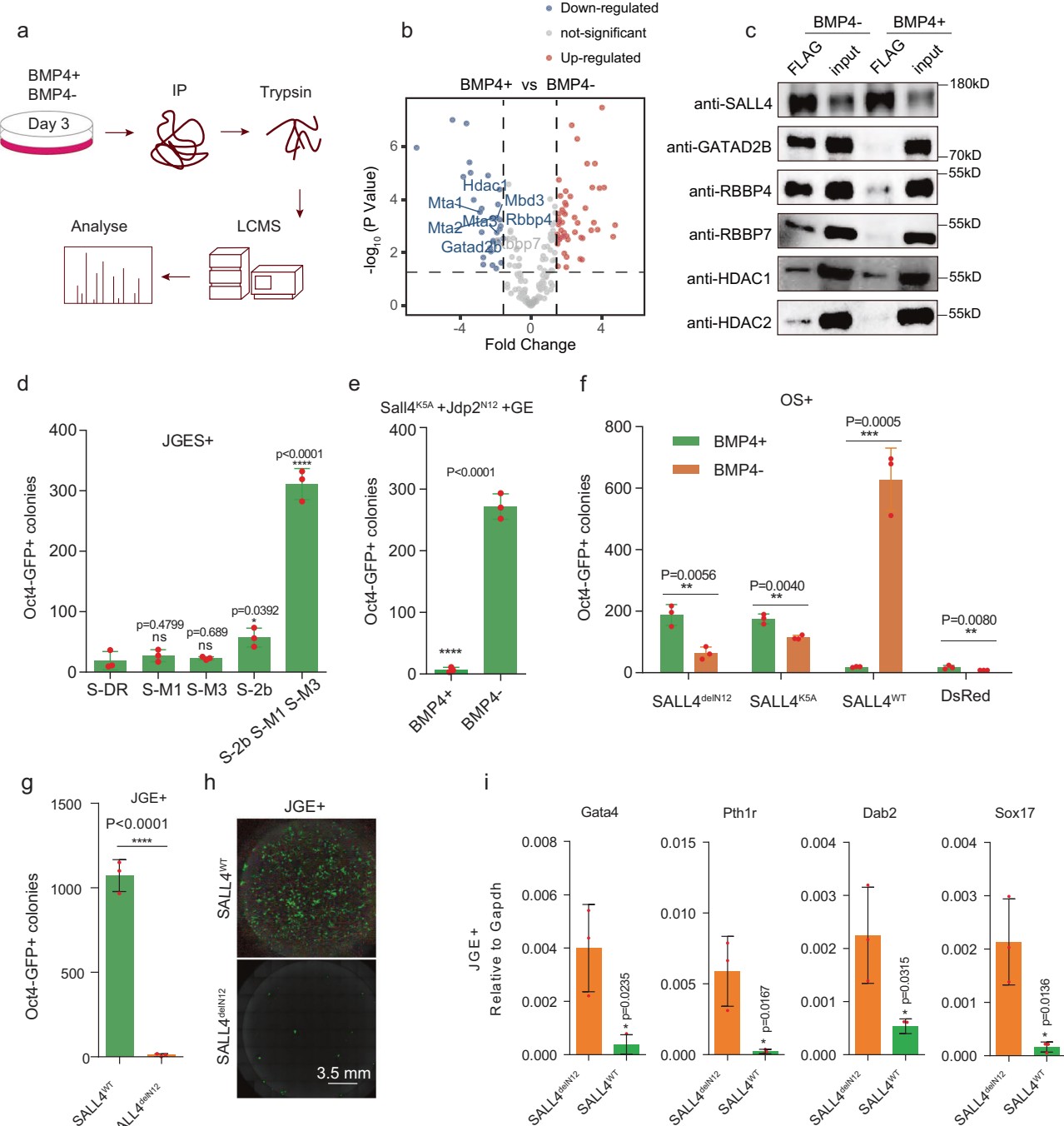

**Fig. 4 | BMP4 dissociates SALL4 from NuRD. a** Flow chart shows the enriched proteins of SALL4 pull down followed by LCMS analysis under BMP4+ and BMP4− conditions of JGES reprogramming at day 3. **b** Volcano map shows down-regulated interaction proteins colored in blue, not significant interaction proteins colored in gray, and up-regulated interaction proteins colored in red, NuRD complex subunits were marked in blue, IP-MS experiments were performed in triplicates and a two-sided *t*-test was applied. *P* < 0.05 and fold change = 1.5 were used as thresholds. **c** Western blots shows the results of SALL4 Co-IP on SALL4-NuRD subunits inter-action under BMP4+ and BMP4− conditions at day 3 of JGES reprogramming, Flag beads were used to pull down SALL4 fusion with Flag tag, and interaction proteins from cell lysates, *n* = 3 independent experiments. **d** Histogram shows *Oct4* GFP positive iPS colonies numbers of JGES reprograming plus fusion proteins in Sup-plementary Fig. 4a, data are mean ± s.d., two-sided, unpaired *t*-test; *n* = 3 indepen-dent experiments, \**p* < 0.05, \*\**p* < 0.01, \*\*\**p* < 0.001. **e** Histogram shows *Oct4* GFP

positive iPS colonies numbers of *Sall4*[K5A], *Jdp2*[N12], *Glis1*, *Esrrb* reprogramming under BMP4+ and BMP4− conditions, data are mean ± s.d., two-sided, unpaired *t*-test; *n* = 3 independent experiments, \**p* < 0.05, \*\**p* < 0.01, \*\*\**p* < 0.001. **f** Histogram shows *Oct4* GFP positive iPS colonies numbers of OS plus *Sall4*[delN12], *Sall4*[K5A], *Sall4*[WT], and DsRed reprograming under BMP4+ and BMP4− conditions, data are mean ± s.d., two-sided, unpaired *t*-test; *n* = 3 independent experiments, \**p* < 0.05, \*\**p* < 0.01, \*\*\**p* < 0.001. **g** Bar plot for *Oct4* GFP positive iPS colonies numbers of JGE plus *SALL4*[WT] or *SALL4*[delN12] reprogramming at Day 7, data are mean ± s.d., two-sided, unpaired *t*-test; *n* = 3 independent experiments, \**p* < 0.05, \*\**p* < 0.01, \*\*\**p* < 0.001. **h** Whole well screening photograph of Fig. 4g, scale bar = 3.5 mm. **i** Histograms show the qPCR results of PrE gene relative expression level of every group, *n* = 3 independent experiments, error bars here represent mean with SD.

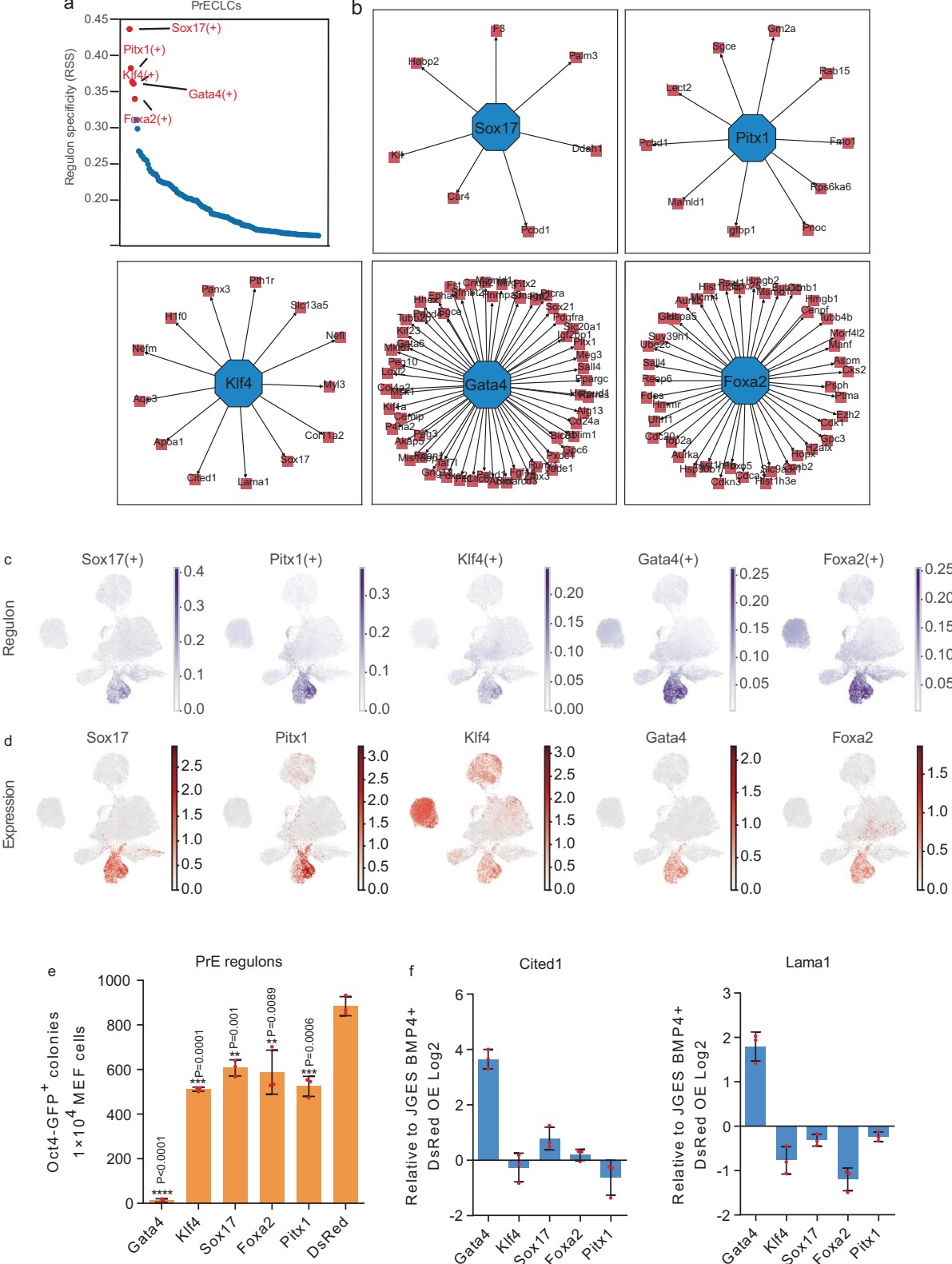

**Fig. 5 | BMP4 activates PrE regulons. a** Plot shows the top 5 regulons of PrE in JGES under BMP4+ condition. **b** Network plots show the transcription factors and their target genes. **c** and **d** UMAP show the regulon scores calculated by SCENIC (**c**) and the expression level (**d**) of transcription factors. **e** Histogram shows *Oct4* GFP positive iPS colonies numbers in different groups, data are mean ± s.d., two-sided, unpaired *t*-test; *n* = 3 independent experiments, *p < 0.05, **p < 0.01, ***p < 0.001. **f** Histograms show the qPCR results of PrE genes relative expression level of every group, *n* = 3 independent experiments, error bars here represent mean with SD.

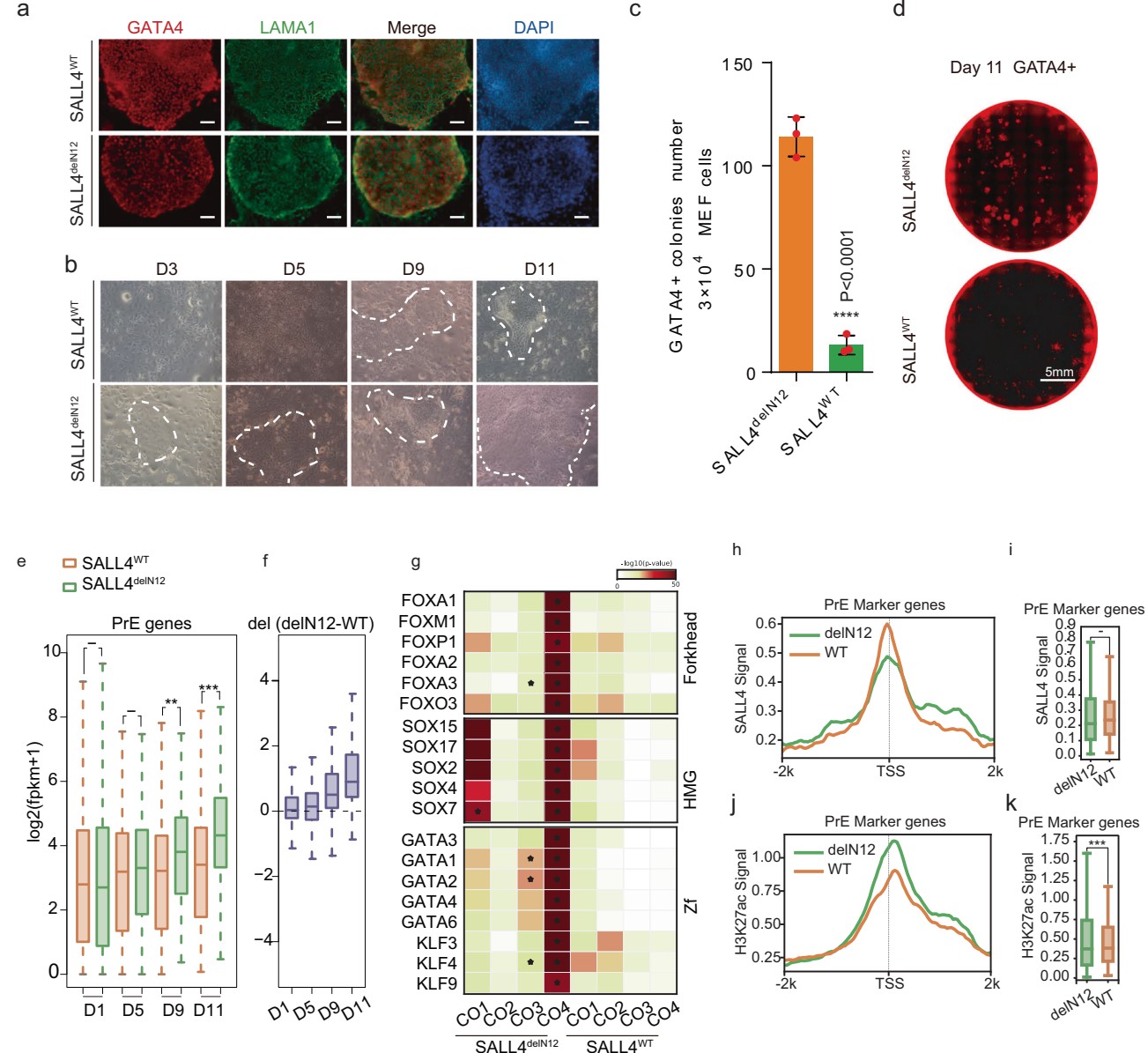

**Fig. 6 | Induction of PrE by SALL4^delN12 alone. a** Immunofluorescence staining shows the iPrE shape clone is GATA4 and LAMA1 dual positive induced by SALL4^WT and SALL4^delN12 at day 11, scale bar = 250 μm. **b** Pictures show the cell morphology change along iPrE induction by SALL4^WT and SALL4^delN12. **c** Bar plot for GATA4 positive iPrE colonies numbers under BMP4− conditions at Day 11, data are mean ± s.d., two-sided, unpaired *t*-test; *n* = 3 independent experiments, *$p < 0.05$, **$p < 0.01$, ***$p < 0.001$. **d** Whole well screening photograph of (**c**), scale bar = 5 mm. **e** and **f** Box plot shows PrE-related gene expression between SALL4^WT and SALL4^delN12 at different time points. A two-tailed Student's *t*-test was performed for comparisons between classes (***$p$-value < 0.001; **$p$-value < 0.01; *$p$-value < 0.05; -$p$-value > 0.05) The middle lines of the boxes indicate the median, the outer edges represent the first and the third quartiles, and the whiskers indicate the 1.5 × interquartile range below the lower quartile and above the upper quartile. **g** Heatmap shows the differentially enriched motifs in the CO groups between SALL4^delN12 and SALL4^WT, as referenced in Supplementary Fig. 6j. **h** Line chart shows the average SALL4 binding signal profile between SALL4^delN12 and SALL4^WT at ±2k of TSS of PrE marker genes. **i** Box plot shows SALL4 binding signal differences between SALL4^delN12 and SALL4^WT at ±2k of TSS of PrE marker genes in (**j**). A two-tailed Student's *t*-test was performed for comparisons between classes. "−" means no significant difference (*p*-value = 0.05), The middle lines of the boxes indicate the median, the outer edges represent the first and the third quartiles, and the whiskers indicate the 1.5 × interquartile range below the lower quartile and above the upper quartile. **j** Line chart shows the average H3K27ac signal profile between SALL4^delN12 and SALL4^WT at ±2k of TSS of PrE marker genes. **k** Box plot shows the H3K27ac signal differences between SALL4^delN12 and SALL4^WT at ±2k of TSS of PrE marker genes in (**i**). A two-tailed Student's *t*-test was performed for comparisons between classes. (***$p$-value < 0.001). The middle lines of the boxes indicate the median, the outer edges represent the first and the third quartiles, and the whiskers indicate the 1.5 × interquartile range below the lower quartile and above the upper quartile.

## Methods

### Mice

*Oct4*-GFP (OG2) transgenic-allele-carrying mice (CBA/CaJ×C57BL/6J) were purchased from The Jackson Laboratory (Mouse strain datasheet: 004654). 129S4/SvJaeJ and ICR mice were purchased from the Beijing Vital River Laboratory. Animals were individually housed under a 12 h light/dark cycle and provided with food and water ad libitum, ambient temperature is about 20−26 °C and humidity is about 40−70%. Our studies followed the guidelines for the Care and Use of Laboratory Animals of the National Institutes of Health, and the protocols were approved by the Committee on the Ethics of Animal Experiments at the Guangzhou Institutes of Biomedicine and Health. OG2 mice and 129Sv/

Jae mice were used to generate E13.5 mouse embryo fibroblast (MEFs) and ICR mice were used to apply donor blastocyst and pseudo-pregnant mice.

## DNA constructs, cell lines and cell culture

pMXs plasmids, pKD plasmids, pLVX plasmids, pB plasmids were used for in vitro over-expression. pMXs (retrovirus vector) were regularly used if not extra mentioned, pKD was used to overexpress rtTA, pLVX carries a TRE3G (Tet-on system) promoter, pB plasmid coupled with pBase plasmid were used to overexpress GFP as indicated. MEFs were obtained from E13.5 mouse embryos regardless of sex by crossing male OG2 mice to 129S4/SvJaeJ/female mice, after removing the integral organs, the tail, the limbs and head, the remaining tissues were cut into small pieces and then digested (0.25% trypsin: 0.05% trypsin = 1:1; GIBCO) for 10 min at 37 °C to a single cell suspension. The isolated MEF cells were seed on 0.2% gelatin (home-made) coated dish, cultured in fibroblast medium: DMEM-high glucose (Hyclone) contains 10% FBS(NTC, SFBE, HK-026), 1% GlutaMAX (GIBCO), 1% sodium pyruvate (GIBCO), 1% NEAA (GIBCO) and 0.1 mM 2-mercaptoethanol (GIBCO). Plat-E cells were cultured in DMEM high-glucose media (Hyclone) supplemented with 10% FBS (NTC, SFBE, HK-026). iPrE cells derived from mouse embryonic fibroblast cells cultured in iCD3: DMEM-high glucose (Hyclone) contains TV, VC, CHIR-99021, bFGF, mLIF, SGC0946, GSK-LSD12HCL, Y27632. 1% sodium pyruvate (GIBCO), 1% non-essential amino acids (GIBCO), 1% GlutaMAX (GIBCO), 0.1 mM 2-mercaptoethanol (GIBCO), N2 (GIBCO), B27 (GIBCO). All the cell lines have been confirmed as mycoplasma contamination-free with the Kit from Lonza (LT07-318).

## iPSCs generation

Plat-E cells and 293T cells were seeded at the concentration of $7.5 \times 10^6 - 8.5 \times 10^6$ Cells per 10-cm dish 12–16 h before transfection uniformly, and cultured in high-glucose DMEM (HyClone, SH30022.01) supplemented with 10% FBS (NTC, SFBE, HK-026) medium. For each 10-cm dish, replacing the Plat-E cells or 293T cells medium with 9 mL fresh 10% FBS firstly, 10 μg DNA was added into 1 ml opti-MEM(GIBCO), mix the liquid immediately after adding 40 μl PEI (1 mg/ml, YEASEN, 40816ES08), for 293T transfection, PSPAX2, PMD2G (6 μg:4 μg) were extra used. After incubating for 10–15 min at room temperature, the mixture should be gently transferred onto the Plat-E cell or 293T cells. Replace the medium with 10 ml 10% FBS within 10–16 h. And then, the retrovirus should be collected twice, 48 and 72 h after transfection, lentivirus should be collected once, 48 h after transfection. The supernatant containing the virus was collected at each time by a syringe and filter through a 0.45 μm filter, 10 ml fresh 10% FBS medium was added to the Plat-E cells after the first collection, the virus can be stored at room temperature for 48 h. Thawing the frozen Passage 1 OG2 MEFs (mouse embryonic fibroblast cells) into a 6 cm dish with 10% FBS medium and cultured in a 5% CO$_2$ incubator when conducting the transfection. Then, split the MEFs into a 24-well plate at $1.5 \times 10^4$ cell density per well before infection. MEF cells should be infected by retrovirus twice and lentivirus once. Mix the virus stock at proper volume (Jdp2:Glis1:Esrrb:Sall4 = 2:1:1:2) and one volume of fresh 10% FBS medium, then mix polybrene at a final concentration of 4 mg/ml, Y27632 at a final concentration of 5 μM before infection. The second virus infection should be conducted 24 h later, lentivirus were infected at the second time. After infection for 2 days, replace the medium with iCD3 or iCD3 plus BMP4 (RD systems, 314-BP-500), change the medium every 24 h, and observe the morphology change. GFP$^+$ clones are captured by living cells station (NIKON, Bio Station CT) and counted by Image-J using particle analysis.

## iPrE generation

MEF cells were seeded onto 24-well plate at $1.5 \times 10^4$ cell density per well, then infected with retrovirus for twice, 24 h each time, iCD3 was used to conduct the generation progress, iPrE will appear gradually. iPrE cells could be digested into single cells by 0.25% trypsin-EDTA, 37 °C, 5 min, along the passages, the extracellular matrix of iPrE cells become more condensed and take a longer digestion time, up to 15 min. iPrE cells could also be cultured in suspension strategy and form a single layer spherical cavity for the cell polarity. Cells could be passaged at a 1:5 ratio, and BMP4 could promote the proliferation of iPrE both in 2D and 3D culture.

## Blastocyst injection and embryo transplantation

The iPrE clones (Oct4-GFP negative, with PrE clone morphology) induced by SALL4$^{WT}$ alone are picked by pipette at day 11, after 3 days the patches are digested into single cells or smaller patches by 0.25% trypsin, and after one or two extra-passage to deplete the non-induced cells, the iPrE cells are ready to be labeled by GFP. MEF cells and iPrE cells are transfected with pB-GFP-2A-puro and pBase (1:1) by lipo 3000, and cultured in iCD3.48 h after transfection, 1 μg/ml puromycin was used to remove the un-transfected cells for extra 48 h until the rest of the cells are all GFP positive. Recover the cells for an extra-passage and then digest them into single cells before injection. Donor blastocysts were isolated by M2 medium from the uterus of female ICR mice 3.5 days after coition, GFP positive cells were injected into the cavity of the blastocyst, 10 cells per embryo, and about 30 embryos were injected per group, one-third of the chimeric embryos were cultured in KOSM in vitro for 48 h to observe MEF cells and iPrE cells proliferation, the rest of the chimeric embryos were transplanted into the uterine horn of the pseudo-pregnant female mouse, about 7–10 chimeric embryos were transplanted into each side of the uterine, chimeric embryos were dissected at E12.5.

## Flow cytometry

Cells were dissociated into single cells using 0.25% trypsin–EDTA and collected after centrifugation at $250 \times g$ for 5 min. After washing with PBS for once, the cell pellet was resuspended with cold PBS containing 0.1% BSA, followed by removing large clumps of cells using a cell strainer (BD Biosciences). The cells were then analyzed by an Accuri C6 flow cytometer (BD Biosciences). The GFP fluorescence cells were detected in the FITC channel. Data analysis was performed using FlowJo v.7.6.1.

## Immunofluorescence

Cells growing on a 96-well dish were washed 3 times with PBS, then fixed with 4% PFA for 0.5 h, after washing 3 times, 10 min per time, by PBS and subsequently penetrated and blocked with 0.2% Triton X-100 and 3% BSA (1:1) for 0.5 h at room temperature. Then, the cells were washed 3 times, 10 min per time, and incubated with primary antibody diluted with 3% BSA for 2 h at room temperature or overnight at 4 °C. After 3 times washing in PBS, the cells were incubated for one hour in second antibodies diluted with 3% BSA at room temperature. After washing 3 times in PBS cells were then incubated in DAPI diluted by PBS for 2 min, plus 3 times washing in PBS. The following antibodies were used in this project: anti-Flag (Sigma Aldrich, F1804, 1:200), anti-SALL4(abcam, ab29112 1:200), anti-GATA4 (Santa Cruz Biotechnology, sc-25310, 1:200), anti-LAMA1(abcam, ab11575 1:200)

## Co-immunoprecipitation and western blot

To perform co-immunoprecipitation, cells were digested with 0.25% trypsin and washed 3 times in PBS, whole cell extracts were prepared using lysis buffer (50 mM Tris pH 7.4, 200 mM NaCl, 10% Glycerol, 1% NP40, 1 mM EDTA) with freshly added 1× Complete Protease inhibitors (Sigma, 1187358001) and 1% PMSF, incubated for 20 min on ice and then 1 h at 4 °C on a rotation wheel. Soluble cell lysates were collected after maximum speed centrifugation at 4 °C for 15 min, part of the supernatant was kept as input, the remaining supernatant was incubated with anti-FLAG beads, DYKDDDDK (Thermo Fisher, A36797)

overnight at 4 °C on a rotation wheel. Beads were then washed with wash buffer (50 mM Tris pH 7.4, 200 mM NaCl, 10% Glycerol, 0.01% NP40, 1 mM EDTA) 10 times by inverting the tube on ice. After complete removal of the cell wash buffer, immunoprecipitated proteins with FLAG beads were boiled at a 100 °C incubator in loading buffer (4% SDS, 10% 2-Mercaptoethanol, 20% Glycerol, 0.004% Bromophenol blue, 0.125 M Tris Ph 6.8) for 10 min, Whole protein extract was stored at −80 °C and avoid freeze and thaw cycle. To perform western blot, input or IP extract was analyzed by SDS−PAGE and then transferred to PVDF membrane (Millipore, 0.45 µm). After incubation with indicated antibodies, the membrane was exposed to X film. NuRD Complex Antibody Sampler Kit (CST, 8349T), anti-GATAD2B (Abcam, ab224391), anti-RBBP4 (Novusbio NB500-123), anti-FLAG (Sigma Aldrich, F1804 1:1000), anti-SALL4 (Abcam, ab29112), were used.

### Processing of scRNA-sequencing data
The FASTQ files of single-cell libraries were generated from Illumina NovaSeq. The clean FASTQ files were aligned to the Mm10 genome with mouse gene annotation of Gencode vM21 version by STARsolo function of STAR (2.7.6a)[48]. Low-quality cells were filtered out by the number of unique molecular identifiers (UMIs) and total counts following the pipeline of Python package Scanpy[49]. Gene regulatory network (GRN) analysis: We performed GRN analysis using pySCENIC[50]. We obtained a regulon score for all cells of each transcription factor. The importance of transcription factors for each cell type was ordered by normalized enrichment score.

### Bulk RNA-seq and data analysis
The RNA-seq reads were trimmed using Trim Galore (v0.6.4)[51,52] and then mapped to the mm10 reference genome with HISAT2 (v2.2.1)[53], and StringTie (v2.2.1)[54] was used to quantify the transcription level of each gene in each sample into fragments per kilobase of exon model per million mapped reads (FPKM). GFOLD (v1.1.4)[55] was used to perform differential expression analysis between conditions. The differentially expressed genes were identified with a gfold value >0.5 or less than −0.5.

### ATAC-seq and data analysis
The ATAC-seq reads were trimmed by Trim Galore (v0.6.4) and then mapped to the mm10 reference genome using bowtie2 (v2.4.5)[56], and SAMtools (v1.16.1)[57] was used to remove the unpaired, low sequencing quality (mapq < 30) and the mitochondrial DNA mapped reads in the total mapped reads. The reads that lengths <50 base pairs (bp) were isolated for subsequent analysis. In order to make the data comparable between different sequencing depths, the signals were normalized to one million reads for each sample, and the values were further compressed into a binary format (bigWig) for downstream analysis and data visualization. Peak calling was performed using MACS (v1.4.2)[58] with parameters as follows: -g mm --keep-dup all --nomodel --shiftsize 25.

### Cut&Tag and data analysis
The CUT&Tag reads were trimmed by Trim Galore (v0.6.4) and then mapped to the mm10 reference genome using bowtie2 (v2.4.5). SAMtools (v1.16.1) was used to remove the repetitive, low sequencing quality (mapq < 30) and the mitochondrial DNA mapped reads in the total mapped reads. The values were further compressed into a binary format for downstream analysis and data visualization. Replicates were merged using samtools merge and then peak calling was performed using MACS (v1.4.2) with parameters as follows: -g mm --keep-dup 1 --nomodel --shiftsize 73. The signals were normalized to one million reads for each sample. Promoters were defined as regions ± 2 kb around transcription start sites (TSSs) of genes.

### Motif analysis
Motif scans were performed using HOMER (v4.11.1)[59] against the genome sequence of the given ATAC-seq peaks covered regions (summits ± 25 bp) with the following parameters: -size given -mask. HOMER used a hypergeometric test to determine the motif enrichment and also test the similarity between the motif we identified and known factors. Motifs that have $p$-value $< 10^{-5}$ and enrichment score $> 3$ are presented in the plot.

### Gene Ontology analysis
Functional annotation was performed using the clusterProfiler (v4.6.2)[60]. Gene Ontology terms for each functional cluster were summarized to a representative term, and adjusted $p$-values were plotted to show the significance.

### Reporting summary
Further information on research design is available in the Nature Portfolio Reporting Summary linked to this article.

## Data availability
The scRNA-Seq, bulk RNA-Seq, ATAC-seq and Cut&Tag data have been deposited in the Gene Expression Omnibus database under the accession code GSE242851. The mass spectrometry proteomics data have been deposited to the ProteomeXchange Consortium via the iProX partner repository[61,62]. with the dataset identifier PXD051433. Source data are provided with this paper.

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

## Acknowledgements

We thank the experimental platforms at Westlake University, Guangzhou Institutes of Biomedicine and Health, Chinese Academy of Sciences, we also appreciate the help from all the members of Duanqing Pei's lab, also in Jing Liu's lab and Jiekai Chen's lab. This work was supported by the National Natural Science Foundation of China (92068201, 32200662), the Key R&D Program of Zhejiang (2024SSYS0033, 2024SSYS0034), Yangtze River Delta Sci-Tech Innovation Community Joint Research Project (2022CSJGG1000),

## Author contributions

J.M. designed the project and performed the main experiments; L.L., C.Z., D.L., and C.L. performed the bioinformatic analysis, and X.H. provided basic bioinformatic counsel; L.W., S.F. and Y.H. performed the cell culture experiments; J.L. performed the blastocyst injection experiments, qPCR experiments, and ATAC-sequence experiments; Y.Y. performed the blastocyst injection experiments; T.H. and W.Z. performed the RNA-seq experiment; Y.F. constructed the plasmids; L.G. performed and X.Z. analyzed the IP-MS experiments; J.K. and Y.Q. provided the experimental counsel. J.C. and J.L. established the basic reprogramming protocol; B.W. established the reprogramming system; D.P. supervised and conceived the whole study, wrote the manuscript, and approved the final version. All authors read and approved the final manuscript.

## Competing interests

The authors declare no competing interests.
