## [Peer Review File · Nature Communications]

Cell fate decision by a morphogen-transcription factor-chromatin modifier axisReviewer #1 (Remarks to the Author):

In this study, Ming et al. presented a hypothesis positing that BMP4 signaling impedes the induction of pluripotency by impairing the NURD-SALL4 interaction complex during JGES-mediated reprogramming. Intriguingly, the authors demonstrated that the introduction of BMP4 redirects reprogramming towards a more primitive endoderm (PrE)-like fate. The authors have provided compelling evidence to support these findings, particularly in their immunoprecipitation (IP) experiments, which exhibit a dissociation between SALL4 and NURD upon BMP4 administration. Furthermore, the authors bolstered their hypothesis by rescuing reprogramming through the covalent binding of SALL4 to various NuRD components. Finally, the authors demonstrate that SALL4 overexpression instructs the PrE fate.

However, the paper could benefit from the following suggestions:

1. Elaboration of the BMP4 mechanism: The paper lacks a detailed mechanism explaining how BMP4 guides reprogramming cells towards a more PrE-like fate. Considering that SALL4 itself can induce PrE fates, it suggests potential competitive binding between SALL4 and PrE-related proteins versus SALL4 and NuRD. The exact mechanism by which BMP4 promotes a PrE fate remains unclear beyond the inability to interact with NuRD, which is necessary for pluripotency. It may be beneficial to explore the upregulated proteins identified in the IP-MS experiments to gain deeper insights into the shift toward PrE. Again, this is particularly relevant since SALL4 alone can reprogram cells to a PrE fate.
2. The blastocyst injection experiments require refinement. It is challenging to discern whether the GFP-labeled cells are contributing to the embryo or primitive endoderm since there is no co-staining with specific markers for PrE. Furthermore, the paper lacks essential statistical analysis, information regarding the number of embryos injected, and other pertinent details. It would be beneficial to include a parallel experiment utilizing pluripotent cells for comparison. Additionally, it is unclear how these experiments are performed. Specifically, are purified PrE cells injected? Or is this a heterogeneous population containing both PrE and pluripotent cells?
3. Inhibition of BMP4 signaling: Considering that BMP4 addition was shown to slow down JGES reprogramming, it raises the question of whether inhibiting BMP4 or downstream targets of BMP4 signaling could enhance the efficiency of reprogramming. Investigating this aspect would strengthen and broaden the author's conclusions.
4. Drop-out screen logic: The rationale behind the drop-out screen is somewhat unclear. Instead of a drop-out to identify factors necessary for pluripotent cells, it might be more informative to perform a drop-out screen to pinpoint the factors required for PrE cell formation. This would focus on understanding the specific components necessary for the observed phenotype rather than pluripotency.
5. Exploration of BMP4's effects on reprogramming: The paper briefly touches on the dichotomy between BMP4's effects on OCT4-based and SALL4-based reprogramming, which could offer a novel aspect to the study. Expanding on this aspect might provide additional insights.
6. Methods: The Methods section requires improvement, with several crucial details missing and certain portions being misplaced. Additionally, it is essential to include statistical analysis for the figures, display individual replicates, and provide the Western blot membranes as supplementary figures.

Reviewer #2 (Remarks to the Author):

Reviewer Comments (Round 1: 20210720)

In this manuscript by Ming and colleagues, the authors explore the role of BMP4 in cell fate switching, when reprogrammed using their previous JGES (Jdp2, Glis1, Esrrb and Sall4) over expression system. Using number of Oct4-GFP positive cells as readout, the authors observe that

BMP4 addition reduces the reprogrammed cells both in a dosage and time-dependent manner. Performing scRNA-seq upon JGES induction, BMP4⁻ cells give rise to intermediate state, endothelial and pluripotent cells, while BMP4⁺ gives rise to intermediate, endothelial and PrECLCs (primitive endoderm-like cells), with both reprogramming yielding ~20% of total single-cell pool. These PrECLCs transcriptomically are shown to be similar to published primitive endoderm (PrE) cells from E4.5-5.5 embryos. The authors subsequently perform a JGES dropout reprogramming screen with and without BMP4, ascertaining ability of BMP4 with Oct4 to reprogram MEFs, and with Sall4 acting antagonistically to BMP4. Using IP-MS, the authors observe perturbed SALL4-NuRD interaction when treating with BMP, and find that BMP4 blocks iPSC reprogramming via disruption of Sall4-NuRD cooperation (N-terminal 12AAs of SALL4). Lastly, the authors show that Sall4DeIN12 alone can reset MEFs to induced extra-embryonic state, faster than Sall4WT; and perform blastocyst injections for iPSC-GFP for extra-embryonic intergration.

The study and results are indeed interesting, the ms is very well written and supported by suitable experiments and analysis. However, the insights from the paper provide marginal gains and mainly improve on authors previous findings.

The reviewer shares a few major and minor concerns stated below.

Comments

- In figure 2A, it was unclear why the authors chose the BMP4 induction after 2 days?
- Figure 1d, highlights that number of Oct4-GFP⁺ colonies are same as un-induced (row 1 vs row 3). Figure 1D-E suggest that BMP4 induction in Days0-3 may not be quite consequential.
- In Figure 2B-C, It is unclear whether there are no pluripotent cells from BMP4⁺ population.
- Additionally a subset of BMP4⁺ are shared with MEF cluster (Pink cells in yellow cluster, Fig2B) even after 7 days, are these non-reprogrammed cells or different state (similar cells are not observed in BMP4⁺ population).
 - o Also look very diff from ESCs.
- The authors report that PrECLCs are PrE cells based on bulk markers and single-cell comparison. The PrE markers are non-zero in BMP4^{-ve} cells? What is the explanation for this? It would be useful to visualise other PrE markers at single-cell level in Fig S2D, to see if the distinction between BMP4⁺ to PrE/PrCLC and BMP4⁻ to pluripotent.
- The reviewer feels that correlation values could be plotted (instead of high low) to appreciate magnitude of similarity.
- In Fig 3b (-sall4) and 3d (oct4, -sall4), the authors should comment on the target gene sets and their activation (cooperativity vs exclusivity), during Oct4 and BMP4 mediated reprogramming. Which sites are unique vs co-bound with Sall4, and regulated?
- In Fig5, the authors should describe number/percentage of PrECLC cells between WT and Sall4DeIN12. Which other cell types emerge during reprogramming (pluripotent, endoderm, intermediate and percentages)? Since the FPKM/counts are non-zero; the cell-fate switch is likely to be non-binary decision.
- Similarly, statistics of Fig5DF and EG could be provided. What percentage of cells of the total were GFP positive.

Minor

- The introduction seems brief and generalised.
- In the discussion, the authors do not comment whether BMP4 mediated cell fate changes are due to JGES TF cooperativity vs exclusivity in action?
- The authors could double check the scalebar between both panels in Fig5c scale. In the print/web version, lower panel seems more magnified.
- What duration and timing of BMP4 induction is most crucial to fate specification in JGES reprogramming? The authors perform a titration and dosage curve, but choose 1 condition for all experiments.

Point-by-point response to the reviewers:

Dear editor and reviewers,

We sincerely thank you for considering and reviewing our work. We especially appreciate the constructive comments that helped us to improve the manuscript through experiments. As suggested by all the reviewers, we performed the relevant experiments and performed literature research to answer all the questions. As such, we sincerely believe that the manuscript has been improved through careful revision as detailed by point-by-point responses below. We hope that your comments have been addressed in full and we obviously welcome any additional ones as well. Please note that the responses are marked with blue color in the text, the references are list at the end of the text, all Figures mentioned in this text have been up-dated in the new version.

Main changes and additions made to Figures.

- 1: Figure 1d: Day 7 to Day 0 is changed to Day 0 to Day 7 from left to right;
- 2: Supplement Figure 1a, 1b, Supplement Figure 2a, 2b, Figure 3i, 3j, Figure 4g, 4h, 4i are added;
- 3: Supplement Figure 2e, 2f, 2g are updated;
- 4: Figure 3a, 3b, 3d, 3e are re-organized;
- 5: Supplement Figure 3a, 3b, 3c, 3d are added;
- 6: Figure 5 and Supplement Figure 5 are new data, previous figure 5 and Supplement Figure 5 are change to Figure 6 and Supplement Figure 6;
- 7: Previous Figure 5i is deleted;
- 8: Previous Figure 5h is changed to Figure 6i;
- 9: Previous Figure 5d, 5e, 5f, 5g are changed to Supplement Figure 6e, 6f, 6g, 6h;
- 10: Figure 6c, 6d, 6e, 6f, 6g, 6h, 6i, 6j, 6k, Supplement Figure 6i, 6j are added.

Reviewer #1 (Remarks to the Author):

In this study, Ming et al. presented a hypothesis positing that BMP4 signaling impedes the induction of pluripotency by impairing the NURD-SALL4 interaction complex during JGES-mediated reprogramming. Intriguingly, the authors demonstrated that the introduction of BMP4 redirects reprogramming towards a more primitive endoderm (PrE)-like fate. The authors have provided compelling evidence to support these findings, particularly in their immunoprecipitation (IP) experiments, which exhibit a dissociation between SALL4 and NURD upon BMP4 administration. Furthermore, the authors bolstered their hypothesis by rescuing reprogramming through the covalent binding of SALL4 to various NuRD components. Finally, the authors demonstrate that SALL4 overexpression instructs the PrE fate.

Response:

We appreciate the reviewer's insightful comments for our work. As reprogramming as a technology has become rather routine and so many nice works have been published over the past 15 years, we branched out into areas far less popular, but perhaps more interesting. One of such areas is the BMP4-Sall4-NuRD axis as detailed in this paper. Beyond the pathways we delineated, we also find, with a bit surprise, that we can generate PrE or PE, the sister fate of epiblasts. So, we decided to look into this further with results we written up for this manuscript. Much needs to be done as pointed out by the reviewer, but we

were encouraged that we have made a good start. So, we especially appreciate the encouraging comments from the reviewer.

However, the paper could benefit from the following suggestions:

1. Elaboration of the BMP4 mechanism: The paper lacks a detailed mechanism explaining how BMP4 guides reprogramming cells towards a more PrE-like fate. Considering that SALL4 itself can induce PrE fates, it suggests potential competitive binding between SALL4 and PrE-related proteins versus SALL4 and NuRD. The exact mechanism by which BMP4 promotes a PrE fate remains unclear beyond the inability to interact with NuRD, which is necessary for pluripotency. It may be beneficial to explore the upregulated proteins identified in the IP-MS experiments to gain deeper insights into the shift toward PrE. Again, this is particularly relevant since SALL4 alone can reprogram cells to a PrE fate.

Response:

We would like to thank the reviewer for pointing out our deficiencies concerning PrE fate control. As we pointed out above, we are perhaps a few labs in the world remaining interested in reprogramming. This is our second paper on the JGES reprogramming system following our initial 7F system paper. Unfortunately, we have yet to see the rest of the field paying attention to this reprogramming system, so the value of this system to the entire stem cell field remains near zero. We thought that we can improve this by showing interesting mechanisms and previously unknown paths. So, we devoted most of our energy to the BMP4-Sall4-NuRD axis, neglected to our detriment the PrE aspect.

As we have learnt since we realised that PrE is the fate diverted by BMPs, PrE is fascinating, but not well described in the literature, to our knowledge, there are only a handful papers describing its derivation, not on the mechanism aspect¹. So, we appreciate the comments from the reviewer and performed experiments to address them as detailed below.

First, we appreciate the insightful suggestion to explore the IP-MS experiments for “potential competitive binding between SALL4 and PrE-related proteins versus SALL4 and NuRD” on BMP4 mechanism study, we look into our IP-MS data and PrE related proteins, the upregulated proteins identified in the SALL4 IP-MS experiments after BMP4 treatment, FC=1.5, pValue=0.05, the same as Fig 4b in the manuscript, 67 proteins appear, at the meantime, We plotted a Venn diagram between the 67 proteins and 142 genes specific expressed in the PrECLCs cluster in our scRNA-seq data Slc9a3r1 turns out to be the only overlapped protein(a, b, c), unfortunately, Slc9a3r1 overexpression can't mimic BMP4's inhibition effect on JGES reprogramming under BMP4-(d, e), Slc9a3r1 KD can't rescue BMP4 induced reprogramming decreasing either(f, g), thus, Slc9a3r1 may not be the critical protein based on these results. Slc9a3r1 is reported to be a HDAC responsive gene, it is a solute-carrier protein and activated upon inhibition or complete loss of HDAC function², this is in agreement with our main conclusion that BMP4 signals to dissociate SALL4-NuRD interaction. Slc9a3r1 is also reported to be required for optimal bone density and bone homeostasis^{3,4} which could be another explanation for the Slc9a3r1 activation by bone morphogenetic protein 4, BMP4.

To gain deeper insights into the shift toward PrE, we look into PrE related and BMP4 induced transcriptome regulation, for SALL4-NuRD axis is to remodel transcriptional profile by altering H3K27ac level, by regulon analyzing on scRNA-sequencing data and additional experiments, *Gata4* turns out to be the key regulon to guide JGES reprogramming to PrE cell fate, BMP4 and *SALL4*^{delN12} can also elevate *Gata4* expression level. BMP4 mechanism on PrE cell fate induction can be explained as below: BMP4 dissociates SALL4-NuRD interaction leads to PrE cell fate formation by up regulate PrE key regulon *Gata4* expression. We also add these data into Figure 5.

a: Venn diagram shows the shared protein between PrE genes and SALL4 up-regulated proteins after BMP4 treatment; b: Line chart shows *Slc9a3r1* expression pattern in BMP4+ and BMP4- condition; c: Line chart shows *Slc9a3r1* expression pattern in WT-SALL4 and delN12-SALL4; d, e, f, g: Histogram shows Oct4 GFP positive iPS colonies numbers in different group, data are mean \pm s.d., two-sided, unpaired t test; n = 3 independent experiments, *p < 0.05, **p < 0.01, ***p < 0.001.

We then discussed if BMP4 inhibits pluripotent cell fate by dissociating SALL4-NuRD interaction. The function of SALL4-NuRD axis is transcriptional regulation by altering H3K27ac level of the target genes which has been clarified in detail in our last paper⁴, so we look into the transcription regulation of the PrE-like cell fate, so as to explain how this cell fate appears and the underlying mechanism of BMP4. We apply regulon analysis by SCENIC method on the PrECLCs cluster with our scRNA-sequence data to identify the main transcription regulons of PrE related genes; the top 5 regulon factors are *Sox17*, *Pitx1*, *Klf4*, *Gata4* and *Foxa2* (a), UMAP shows the regulon scores calculated by SCENIC (top panel) and the expression level (lower panel) of transcription factors(b), network plot shows the transcription factors and their target genes (c), *Gata4* regulated the largest PrE gene number among the top 5 regulons and it is more restricted to PrECLCs than the other four regulons(b). Over-expression (OE) experiments shows *Gata4* has the strongest inhibition effect on pluripotency induction(d) and promotion effect on PrE formation(e), Supplementary Figure 2b indicates BMP4 can elevate *Gata4* expression, and Knocking-

down *Gata4* in JGES reprogramming with BMP4 treatment can also inhibit PrE fate formation by qPCR experiments (f, g). These data indicate BMP4 guides reprogramming cells towards a more PrE-like fate through activating *Gata4*.

Besides this, we also want to know the relationship between BMP4 activating PrE regulon genes expression and dissociating SALL4-NuRD interaction. We apply *SALL4^{delN12}* and *SALL4^{WT}* with JGE without BMP4 treatment, and show that *SALL4^{delN12}* could mimic BMP4 function both on pluripotency inhibition(h, i) and PrE promotion(j) .Previous study has indicated *Gata4* and other PrE related markers are absent in homozygous mutant *Sall4* cell cultures derived from ICM⁵. In conclusion, BMP4 dissociates SALL4-NuRD interaction leading to PrE cell fate formation by upregulating PrE key regulon *Gata4* expression. We add these data to Figure 5 in the new version.

a: Plot shows the top 5 regulons of PrE; b: UMAP shows the regulon scores calculated by SCENIC (top panel) and the expression level (lower panel) of transcription factors; c: Network plot shows the transcription factors and their target genes; d: Histogram shows Oct4 GFP positive iPS colonies numbers in different group, data are mean \pm s.d., two-sided, unpaired t test; n = 3 independent experiments, *p < 0.05, **p < 0.01, ***p < 0.001; e, f, g: Histograms show the qPCR results of PrE gene relative expression level of every group; h: Histogram shows Oct4 GFP positive iPS colonies numbers in different group, data are mean \pm s.d., two-sided, unpaired t test; n = 3 independent experiments, *p < 0.05, **p < 0.01, ***p < 0.001; i: Pictures show the in situ whole well screening of Oct4 GFP positive clone number of different group, scale bar= 5mm; j: Histograms show the qPCR results of PrE gene relative expression level of every group, data are mean \pm s.d., two-sided, unpaired t test; n = 3 independent experiments, *p < 0.05, **p < 0.01, ***p < 0.001.

2. The blastocyst injection experiments require refinement. It is challenging to discern whether the GFP-labeled cells are contributing to the embryo or primitive endoderm since there is no co-staining with specific markers for PrE. Furthermore, the paper lacks essential statistical analysis, information regarding the number of embryos injected, and other pertinent details. It would be beneficial to include a parallel experiment utilizing pluripotent cells for comparison. Additionally, it is unclear how these experiments are performed. Specifically, are purified PrE cells injected? Or is this a heterogeneous population containing both PrE and pluripotent cells?

Response:

As stated above, we really appreciate these suggestions and added “The iPrE clones induced by SALL4^{WT} alone are picked by pipette at day 11, after 3 days, the patches are digested into single cells or smaller patches by 0.25% trypsin. After one or two extra-passages to deplete the non-induced cells, the iPrE cells

are ready to be labeled by GFP.” to the method part. iPrEs-GFP we used to perform the blastocyst injection experiments should be a heterogeneous population containing both iPrE cells and very rare non-iPrE cells but no pluripotent cells. For iPrE shaped clones are all *Oct4*-GFP negative, there is no comparability between iPrEs and pluripotent cells. Indeed we applied MEF cells marked by GFP as a parallel experiment for the iPrE cells are derived from MEFs, and the results indicate iPrE could incorporate into PrE upon blastocyst injection but not MEFs (Supplementary Figure 6e, 6f).

As previously described in the Figure legend 5g, “ g. Pictures show iPrE-GFP emerge at embryo yolk sac after transplantation. 8 embryos were obtained at E12.5, 5 of which have GFP+ cells in their yolk sac, no GFP+ cells were found in embryo yolk sac of MEFs-GFP group, data not shown.” 5/8 embryos we obtained have GFP+ cells in their yolk sac, and we showed two of them in the previous Figure 5 (Batch 1). In order to replenish the essential statistical analysis, we did the blastocyst injection experiment four more times. Statistical data are shown in the table below, and we also added this table to Supplementary Figure 6i:

Batch	Surrogate Mice	iPrEs Injected Blastocysts	Embryos at E12.5	Embryos With GFP Labeled
1	2	28	8	5
2	1	7	2	2
3	2	12	0	1
4	1	14	2	2
5	2	22	7	3
SUM	8	83	19	12

PrE cells arise around E4.5-E5.5 in the embryo development in vivo, and these cells are PrE markers positive. However, when they take part in the formation of extra-embryonic yolk sac at E12.5, they become multiple cell types, so when we inject our GFP+ iPrE cells into E3.5 embryo cavity, it is difficult to trace which kind of cells they are in the extra-embryonic yolk sac, as it is very difficult to co-stain extra-embryonic yolk sac markers with GFP labeled iPrE cells. This blastocyst injection experiments we performed was used to test the in vivo developmental ability of the iPrE cells, like the previous study tests the in vivo developmental potential of PrESCs (primitive endoderm stem cells) separated from blastocyst⁶. However, the chimeric efficiency of iPrEs is far below PrESCs (Supplementary Figure 6g, 6h), we are still trying to enhance the induction and chimeric efficiency of iPrE cells.

To further determine the cell identity of iPrE, we also performed IF (Fig 6a) to test the PrE markers expression in iPrE clones, and detected retrovirus silencing, a characteristic of mouse stem cells⁷ (Supplementary Figure 6b), iPrE cells show strong proliferation capacity as well(Supplementary Figure 6c). Unlike ESCs, PrESCs can form a single layer cavity under suspension culture, exhibiting polarity characteristics and condensed extracellular matrix⁸, iPrE cells are also capable of this feature(Supplementary Figure 6d) . Together, these iPrE cells have primitive endoderm like characteristics.

3. Inhibition of BMP4 signaling: Considering that BMP4 addition was shown to slow down JGES reprogramming, it raises the question of whether inhibiting BMP4 or downstream targets of BMP4 signaling could enhance the efficiency of reprogramming. Investigating this aspect would strengthen and broaden the author's conclusions.

Response:

We again appreciate very much this insight., BMP4 signaling pathway belongs to the TGF- β super family, of which the SMADs proteins are downstream effectors, *Smad6* and *Smad7* are inhibitory SMADs of TGF- β signaling⁹, *Smad6* is more specific to BMP4 signaling than *Smad7*^{10,11}. We over-expressed *Smad6* and *Smad7* individually in JGES reprogramming with or without BMP4 treatment, and show that both SMADs could markedly rescue the inhibition effect caused by BMP4, the rescue efficiency of *Smad6* is higher than that of *Smad7* which is in accord with the inhibition preference between the two genes. We also add these results into Supplementary figure 1.

a, Histogram shows Oct4 GFP positive iPS colonies numbers in different group, data are mean \pm s.d., two-sided, unpaired t test; n = 3 independent experiments, *p < 0.05, **p < 0.01, ***p < 0.001; b: Pictures show the in situ whole well screening of Oct4 GFP positive clone number of different group, scale bar= 5mm

4. Drop-out screen logic: The rationale behind the drop-out screen is somewhat unclear. Instead of a drop-out to identify factors necessary for pluripotent cells, it might be more informative to perform a drop-out screen to pinpoint the factors required for PrE cell formation. This would focus on understanding the specific components necessary for the observed phenotype rather than pluripotency.

Response:

We welcome the opportunity to clarify the logic for the drop-out screen experiments. We clarified this part in the manuscript and reorganized the Figures to a better reflect of our drop-out logic.

We also performed extra experiments to pinpoint the requirement of factors for PrE cell formation. When we dropout every single factor of JGES under BMP4+ condition, the qPCR results show that except for *Jdp2*'s dropout which shows a strong promotion of PrE genes, We hypothesized that the other three factor *Glis1*, *Esrrb* and *Sall4* may form an interaction network for PrE formation, To overcome this confounding effect and identify which factor is essential for PrE cell formation, we added the four factors individually into OS reprogramming with BMP4 treatment, and show that that *Sall4* is the only factor to increase PrE gene expression level, consistent with notion that BMP4 functions to divert cell fate from pluripotency to PrE. We also add this data into Figure 3i, 3j.

a, b : Histograms show the qPCR results of PrE gene relative expression level of every group, n = 3 independent experiments.

5. Exploration of BMP4's effects on reprogramming: The paper briefly touches on the dichotomy between BMP4's effects on OCT4-based and SALL4-based reprogramming, which could offer a novel aspect to the study. Expanding on this aspect might provide additional insights.

Response:

Indeed, the most surprising result is the opposing effect of BMP4 on Oct4- and Sall4- centered reprogramming. We have shown, in our view quite convincingly, that BMP4 can either be a positive or negative factor in reprogramming depending on the factors used, perhaps reflecting the diversity of paths towards pluripotency at present¹². Given the complexity of BMP4 mediated pathways, it remains quite surprising that it can exert this unexpected role in cell fate control in vitro¹³.

Apart from BMP4 being a reprogramming enhancer, in Oct4-based reprogramming¹⁴⁻¹⁶, NuRD in general has been regarded as a reprogramming barrier in reprogramming system reported so far^{17,18}. However, it is indeed surprising as described above that both BMP4 and NuRD behave quite differently in JGES reprogramming system. As we reported in our last paper, Sall4 is important to drive JGES reprogramming, and that NuRD is the most important interaction partner of SALL4. In this paper, we reveal further surprise that BMP4 is inhibitory for JGES reprogramming. The BMP4-Sall4-NuRD axis remains functional even in the context of Oct4-based reprogramming as shown in Figure 3e, 3g, 3h, which strongly suggests that this axis is a robust pathway not recognized until our work.

However, we feel that our results are clear enough that we do not need to expand excessively. However, the implication remains quite clearly that cell fate control is context dependent and should be examined as carefully as possible as we demonstrated here.

6. Methods: The Methods section requires improvement, with several crucial details missing and certain portions being misplaced. Additionally, it is essential to include statistical analysis for the figures, display individual replicates, and provide the Western blot membranes as supplementary figures.

Response:

We really appreciate this suggestion. We have tried to improved our method details and appended statistical analysis of the figures and manuscript, also added the Western blot membranes into the Statistics Source Data.

Reviewer #2 (Remarks to the Author):

Reviewer Comments (Round 1: 20210720)

In this manuscript by Ming and colleagues, the authors explore the role of BMP4 in cell fate switching, when reprogrammed using their previous JGES (Jdp2, Glis1, Esrrb and Sall4) over expression system. Using number of Oct4-GFP positive cells as readout, the authors observe that BMP4 addition reduces the reprogrammed cells both in a dosage and time-dependent manner. Performing scRNA-seq upon JGES induction, BMP4- cells give rise to intermediate state, endothelial and pluripotent cells, while BMP4+ gives rise to intermediate, endothelial and PrECLCs (primitive endoderm-like cells), with both reprogramming yielding ~20% of total single-cell pool. These PrECLCs transcriptomically are show to be similar to published primitive endoderm (PrE) cells from E4.5-5.5embryos. The authors subsequently perform a JGES dropout reprogramming screen with and without BMP4, ascertaining ability of BMP4 with Oct4 to reprogram MEFs, and with Sall4 acting antagonistically to BMP4. Using IP-MS, the authors observe perturbed SALL4-NuRD interaction when treating with BMP, an find that BMP4 blocks iPSC reprogramming via disruption of Sall4-NuRD cooperation (N-terminal 12AAs of SALL4). Lastly, the authors show that Sall4DeIN12 alone can reset MEFs to induced extra-embryonic state, faster than Sall4WT; and perform blastocyst injections for iPrE-GFP for extra-embryonic intergration.

The study and results are indeed interesting, the ms is very well written and supported by suitable experiments and analysis. However, the insights from the paper provide marginal gains and mainly improve on authors previous findings.

The reviewer shares a few major and minor concerns stated below.

Comments

- In figure 2A, it was unclear why the authors chose the BMP4 induction after 2 days?

Response:

We apologize for this confusion. We normally allowed the cells to be infected with the reprogramming factors for 2 days based on our previous optimizations.. In figure 2A, the “day 2” means 2 days post transfection, when we start the induction process under BMP4+ or BMP4-, “day 7” means 7 days post transfection we start the scRNA-sequencing process, we have changed “day 2” into “transfection for 2 days” in Figure 2a.

Figure 1d, highlights that number of Oct4-GFP+ve colonies are same as un-induced (row 1 vs row 3). Figure 1D-E suggest that BMP4 induction in Days0-3 may not be quite consequential.

Response:

Again, we appreciate this opportunity to clarify this confusion. The schema chart in Figure 1d shows the number of *Oct4*-GFP+ clones at different time windows. The condition with BMP4 treatment are colored in blue, and without BMP4 treatment are colored in gray. Row 1 is without BMP4 treatment and row 3 means with BMP4 treatment from day 4 to day 7. Based on these results, we show that BMP4 doesn't inhibit JGES reprogramming efficiency from day 4-day 7. Figure 1d exhibits that BMP4's inhibition effect only occurs in the first three days. We changed Day 7 to Day 0 labels to Day 0 to Day 7 from left to right to make the figure more intuitive.

In our previously paper, SALL4-NuRD axis is important to silence somatic-related genes which is necessary in the early stage in reprogramming¹⁹, Figure 1d also shows that BMP4 dissociates SALL4-NuRD interaction. Figure 1e and f show that the reprogramming efficiency of JGES under different BMP4 dosages from day 0 to day 7. In order to eliminate the high dosage inhibition effect, we applied different BMP4 dosage in JGES, the results show that 1ng/ml BMP4 is able to inhibit half of Oct4-GFP+ clones, demonstrating that BMP4's inhibition function on JGES reprogramming is dosage dependent.

- In Figure 2B-C, It is unclear whether there no no pluripotent cells from BMP4+ population.

Response:

Indeed, there is no pluripotent cells in BMP+ population with JGES reprogramming. In Figure 2b-c, BMP4+ populations are colored in red in Figure 2b, pluripotent cell group is colored in red in Figure 2c. When align together carefully, we can find there are very rare BMP4+ populations located in pluripotent group. To make this clear, we added new charts to show BMP4+ and BMP4- populations in different cell types in Figure 2c. We added these charts to Supplementary Figure 2a-c.

a: UMAP shows the cells distribution of BMP4+ and BMP4- conditions; b: Stacked barplot shows the cell proportion in each cell type.

- Additionally a subset of BMP4+ are shared with MEF cluster (Pink cells in yellow cluster, Fig2B) even after 7 days, are these non-reprogrammed cells or different state (similar cells are not observed in BMP4+ population). Also look very diff from ESCs.

Response:

We indeed noticed the subset of BMP4+ which shared the same cluster with MEF in Fig 2b. However, we don't think they are non-reprogrammed MEF cells, but at a different state from MEF cells as well as ESCs, based on the following reason.

Firstly, we analysed the gene expression pattern between the two subsets, there are 90 genes commonly expressed in both subsets, and 51 genes specific expressed in MEFs, 157 genes specifically in BMP4+ (a). GO analysis show the genes commonly expressed are related to wound healing, collagen fibril organization, positive regulation of cell-substrate adhesion and extracellular structure organization. Genes, specifically expressed in MEF are related to extracellular matrix assembly, ossification, bone mineralization and biomineral tissue development. Finally, genes, specifically expressed in BMP4+ are related to regulation of protein stability, regulation of epithelial cell migration, Wnt signaling pathway and cell-cell signaling by Wnt (b). MEF subset specific express *Atp5o*, *Srp54b*, *Aspn* etc, but BMP4+ subset specific express *Ins2*, *Pgk1*, *ApoE* etc (c). In terms of gene quantity ,52.7% genes expressed in BMP4+ subset are different from MEF cells, in terms of gene function, BMP4+ subset is also very different from MEF cells, and so BMP4+ subset within the MEF cluster should be a different state from MEF cells. So, based on these results, we believe that these cells are different from MEFs, but at different stages as described.

a: Heatmap shows the common and different features between MEF and BMP4+ cells in cluster 2; b: Bar plots show the top 5 GO terms of gene sets; c: UMAP shows the expression of selected marker genes

- The authors report that PrECLCs are PrE cells based on bulk markers and single-cell comparison. The PrE markers are non-zero in BMP4-ve cells? What is the explanation for this?

Response:

Based on single-cell comparison, bulk markers, IF experiments, indeed we have detected a subset of cells from JGES reprogramming with BMP4 treatment are PrE cell-like cells. The non-zero PrE markers in the BMP4- group is because there are a background level ~ 0.16% PrECLCs in BMP4- cluster from our scRNA-seq data shown in Figure 2f colored in green. This is because SALL4 alone could reset MEF cells to PrECLCs state at very low efficiency without BMP4 treatment as shown in Figure 6a. We think PrECLCs arise from JGES reprogramming as an alternative fate and indeed over-expressing PrE marker genes in JGES could decrease *Oct4*-GFP+ clone number as shown in Figure 2h. Therefore, JGES reprogramming generates a background level of PrE, making up markers as non-zero in BMP4- group.

- It would be useful to visualise other PrE markers at single-cell level in Fig S2D, to see if the distinction between BMP4+ to PrE/PrCLC and BMP4- to pluripotent. The reviewer feels that correlation values could be plotted (instead of high low) to appreciate magnitude of similarity

Response:

We appreciate this advice. To better compare the states of BMP4+ to PrE/PrECLC and BMP4- to pluripotent cells, we used a new dataset²⁰ whose quality is superior to the one previously used²¹, PrECLCs of BMP4+ can co-localize with E4.5-5.5 PrE, pluripotency cells of BMP4- can co-localize with E4.5-5.5 epiblast (a, b). We also visualized additional pluripotency and PrE markers among the two clusters at single cell level, and observed shared gene expression as well as differential gene expression between reprogramming cells and in vivo cells (c, d, e, f). We also used the CCA algorithm to integrate reprogramming data with mouse embryo data of E4.5-E5.5²². The results show that PrECLCs are similar to in vivo PrE cells, with a Pearson correlation coefficient of 0.88, while Pluripotent cells are similar to in vivo Epiblast cells, with a correlation coefficient of 0.86 (g). We updated it to Supplementary Figure 2e, 2f, 2g in the new manuscript version.

a, b: UMAP shows the integration of PrECLCs/Pluripotent cells' data in this study and Epiblast/PrE data of Sala2019 in vivo;
 c, d, e, f : UMAP shows the expression of marker genes at single cell level; g: Correlation heatmap of reprogramming cells and developmental cells.

- In Fig 3b (-sall4) and 3d (oct4, -sall4), the authors should comment on the target gene sets and their activation (cooperativity vs exclusivity), during Oct4 and BMP4 mediated reprogramming. Which sites are unique vs co-bound with Sall4, and regulated?

Response:

We appreciate this insightful suggestion. To answer this question, we performed bulk RNA-sequence on JGE vs JGES and, JGEO vs JGESO under BMP4+ and BMP4- condition, and evaluated BMP4's function on pluripotency gene sites and PrE related gene sites with or without *Sall4*. The results show that BMP4 can decrease pluripotency genes expression dramatically when *Sall4* exists both in JGE vs JGES and JGEO vs JGESO (a b); BMP4 can decrease pluripotency genes expression dramatically when *Sall4* exists both in JGE vs JGES and JGEO vs JGESO, BMP4 can also increase PrE genes expression dramatically when *Sall4* presents in JGE vs JGES, in JGEO vs JGESO, BMP4 could also upgrade PrE genes expression slightly(c, d). All these data indicate that BMP4 functions to inhibit pluripotency gene and promote PrE gene sites when *Sall4* present. We added the results to Supplementary Figure 3a, 3b, 3c, 3d.

a, Box plot shows the expression levels of pluripotency genes between conditions with and without BMP4 treatment at day 7 during JGE, JGES, JGEO, JGESO reprogramming; b, Box plot shows the fold changes in the expression of pluripotency genes between conditions with and without BMP4 treatment at day 7, during the JGE, JGES, JGEO, JGESO reprogramming, represented on a logarithmic scale; c, Box plot shows the expression levels of PrE related genes between conditions with and without BMP4 treatment at day 7 during JGE, JGES, JGEO, JGESO reprogramming; d, Box plot shows the fold changes in the expression of PrE related genes between conditions with and without BMP4 treatment at day 7, during the JGE, JGES, JGEO, JGESO reprogramming, represented on a logarithmic scale.

- In Fig5, the authors should describe number/percentage of PrECLC cells between WT and Sall4DeIN12. Which other cell types emerge during reprogramming (pluripotent, endoderm, intermediate and percentages. Since the FPKM/counts are non-zero; the cell-fate switch is likely to be non-binary decision.

Response:

Indeed, we agree with the reviewer's suggestion. Here we show the IF results of GATA4+ clone number (a) and whole well screening pictures (b) between WT and delIN12-SALL4 PrE induction at day 11, delIN12-SALL4 group is much higher than WT-SALL4. We agree with the opinion, the cell-fate switch is not a binary decision between BMP4+ and BMP4- group of JGES reprogramming, not even in WT-SALL4 and delIN12-SALL4 single factor induced reprogramming, PrECLCs cluster is not the only cell group which obtain a proportion increase after BMP4 treatment, however, PrECLCs is a result of BMP4 induced SALL4-NuRD dissociation in JGES reprogramming, SALL4-NuRD (delIN12-SALL4) dissociation leads to pluripotency inhibition (c, d) and PrE cell fate promotion (e).

In the single factor induced reprogramming, we checked the expression pattern of pluripotency, endoderm and intermediate related marker genes, based on bulk RNA-sequence data, unlike the expression pattern of PrE genes, delN12-SALL4 doesn't show an improvement effect on these cell lineage (f, g, h). Although PrE cell fate may not be the only direction of delN12-induced MEF cells, additional function will be discussed further in our subsequent research.

a, Histogram shows IF results of GATA4 positive iPrE colonies number in different group, data are mean \pm s.d., two-sided, unpaired t test; n = 3 independent experiments, *p < 0.05, **p < 0.01, ***p < 0.001; b: Pictures show the in situ whole well screening of GATA4 positive iPrE colonies number of different group, scale bar= 10mm; c, Histogram shows Oct4 GFP positive iPS colonies number in different group, data are mean \pm s.d., two-sided, unpaired t test; n = 3 independent experiments, *p < 0.05, **p < 0.01, ***p < 0.001; d: Pictures show the in situ whole well screening of Oct4 GFP positive clone number of different group, scale bar= 5mm; e, Histograms show the qPCR results of PrE gene relative expression level of every group, n = 3 independent experiments; f, g, h: Line charts show the expression pattern of different cell lineage markers between SALL4^{WT} and SALL4^{delN12}.

- Similarly, statistics of Fig5DF and EG could be provided. What percentage of cells of the total were GFP positive.

Response:

We again appreciate this suggestion. However it is really hard to quantify the exact GFP positive cell percentage in the whole yolk sac at E12.5 chimeric embryos. We performed the blastocyst injection experiments to test iPrE's in vivo developmental ability. In previous studies, in vivo developmental potential of PrESCs (primitive endoderm stem cells) separated from blastocyst are also tested by this way ⁶. However, the chimeric efficiency of iPrEs is indeed lower than PrESCs (Supplementary Figure 6g, 6h). Additionally, it is very hard to digest the GFP positive cells from the yolk sac for it has very condense extra-cellular matrix and complicated tissue structure, our previous study also didn't show the exact cell percentage of the chimerism cells, because of these limitations. This technological difficulties are formidable enough to provide a more exact assessment. Nevertheless, we are trying to enhance the induction and chimeric efficiency of iPrE cells.

Minor

- The introduction seems brief and generalised.

Response:

Highly appreciate your suggestion, we improved the introduction part of this manuscript. By providing more relevant background on PrE, BMP and also early cell fate decisions.

- In the **discussion**, the authors do not comment whether BMP4 mediated cell fate changes are due to JGES TF cooperativity vs exclusivity in action?

Response:

Reviewer #1 also asked a similar question. We performed the JGES TF dropout experiments and added the TF one by one into OS-derived reprogramming with BMP4 treatment again, after that we check the PrE genes expression among them by qPCR, the results indicate that except Jdp2 dropout has a strong promotion of PrE genes expression, the other three factor dropout all inhibit PrE cell fate, *Glis1*, *Esrrb* and *Sall4* may form an cooperativity interaction network for PrE formation and Jdp2 is a antagonists in PrE cell fate formation (a), however, in OS reprogramming, *Sall4* is the only factor to increase PrE gene expression level (b), *Sall4* is important in PrE cell fate formation both in JGES and OS system under BMP4 treatment. We revised the discussion by focusing more on the effect of SALL4 more and its impact with BMP4 treatment. We are not sure about cooperativity or exclusivity. It is clear that BMP4 is targeting Sall4-NuRD for the cell fate decisions.

a, b: Histograms show the qPCR results of PrE gene relative expression level of every group, n = 3 independent experiments.

- The authors could double check the scalebar between both panels in in Fig5c scale. In the print/web version, lower panel seems more magnified

Response:

Thanks for this suggestion and sorry for the mistake we made, we change the picture we used in Figure 6a.

- What duration and timing of BMP4 induction is most crucial to fate specification in JGES reprogramming? The authors perform a titration and dosage curve, but choose 1 condition for all experiments.

Response:

According to Figure 1d, e, f, the first three days is the most crucial time window for BMP4 pluripotency inhibition. However, BMP4 has the strongest inhibition on JGES under day0-day7, so we chose this condition. The reason why we perform a titration and dosage curve is to confirm BMP4's inhibition on JGES reprogramming is not due to nonspecific effect. Since 10 ng/ml BMP4 has been proved to be a reprogramming promoter in previously studies^{14,23}, is also an effective inhibition dosage in Figure 1e, so we choose 10ng/ml.

- 1 Graham, S. J. *et al.* BMP signalling regulates the pre-implantation development of extra-embryonic cell lineages in the mouse embryo. *Nat Commun* **5**, 5667, doi:10.1038/ncomms5667 (2014).
- 2 Dudakovic, A. *et al.* Histone deacetylase inhibition destabilizes the multi-potent state of uncommitted adipose-derived mesenchymal stromal cells. *J Cell Physiol* **230**, 52-62, doi:10.1002/jcp.24680 (2015).
- 3 Dudakovic, A. *et al.* Histone deacetylase inhibition promotes osteoblast maturation by altering the histone H4 epigenome and reduces Akt phosphorylation. *J Biol Chem* **288**, 28783-28791, doi:10.1074/jbc.M113.489732 (2013).
- 4 Schroeder, T. M., Nair, A. K., Staggs, R., Lamblin, A. F. & Westendorf, J. J. Gene profile analysis of osteoblast genes differentially regulated by histone deacetylase inhibitors. *BMC Genomics* **8**, 362, doi:10.1186/1471-2164-8-362 (2007).
- 5 Elling, U., Klasen, C., Eisenberger, T., Anlag, K. & Treier, M. Murine inner cell mass-derived lineages depend on Sall4 function. *Proc Natl Acad Sci U S A* **103**, 16319-16324, doi:10.1073/pnas.0607884103 (2006).
- 6 Ohinata, Y. *et al.* Establishment of mouse stem cells that can recapitulate the developmental potential of primitive endoderm. *Science* **375**, 574-578, doi:10.1126/science.aay3325 (2022).
- 7 Schorn, A. J., Gutbrod, M. J., LeBlanc, C. & Martienssen, R. LTR-Retrotransposon Control by tRNA-Derived Small RNAs. *Cell* **170**, 61-71.e11, doi:10.1016/j.cell.2017.06.013 (2017).
- 8 Tosenberger, A. *et al.* A multiscale model of early cell lineage specification including cell division. *NPJ Syst Biol Appl* **3**, 16, doi:10.1038/s41540-017-0017-0 (2017).
- 9 Hanna, A. & Frangogiannis, N. G. The Role of the TGF- β Superfamily in Myocardial Infarction. *Front Cardiovasc Med* **6**, 140, doi:10.3389/fcvm.2019.00140 (2019).
- 10 Choi, K. C. *et al.* Smad6 negatively regulates interleukin 1-receptor-Toll-like receptor signaling through direct interaction with the adaptor Pellino-1. *Nat Immunol* **7**, 1057-1065, doi:10.1038/ni1383 (2006).
- 11 Estrada, K. D., Retting, K. N., Chin, A. M. & Lyons, K. M. Smad6 is essential to limit BMP signaling during cartilage development. *J Bone Miner Res* **26**, 2498-2510, doi:10.1002/jbmr.443 (2011).
- 12 Hanna, J. *et al.* Direct cell reprogramming is a stochastic process amenable to acceleration. *Nature* **462**, 595-601, doi:10.1038/nature08592 (2009).
- 13 Sadlon, T. J., Lewis, I. D. & D'Andrea, R. J. BMP4: its role in development of the hematopoietic system and potential as a hematopoietic growth factor. *Stem Cells* **22**, 457-474, doi:10.1634/stemcells.22-4-457 (2004).
- 14 Chen, J. *et al.* BMPs functionally replace Klf4 and support efficient reprogramming of mouse fibroblasts by Oct4 alone. *Cell Res* **21**, 205-212, doi:10.1038/cr.2010.172 (2011).
- 15 Hayashi, Y. *et al.* BMP-SMAD-ID promotes reprogramming to pluripotency by inhibiting p16/INK4A-dependent senescence. *Proc Natl Acad Sci U S A* **113**, 13057-13062, doi:10.1073/pnas.1603668113 (2016).
- 16 Yu, S. *et al.* BMP4 resets mouse epiblast stem cells to naive pluripotency through ZBTB7A/B-mediated chromatin remodelling. *Nat Cell Biol* **22**, 651-662, doi:10.1038/s41556-020-0516-x (2020).
- 17 Luo, M. *et al.* NuRD blocks reprogramming of mouse somatic cells into pluripotent stem

- cells. *Stem Cells* **31**, 1278-1286, doi:10.1002/stem.1374 (2013).
- 18 Rais, Y. *et al.* Deterministic direct reprogramming of somatic cells to pluripotency. *Nature* **502**, 65-70, doi:10.1038/nature12587 (2013).
- 19 Wang, B. *et al.* The NuRD complex cooperates with SALL4 to orchestrate reprogramming. *Nat Commun* **14**, 2846, doi:10.1038/s41467-023-38543-0 (2023).
- 20 Pijuan-Sala, B. *et al.* A single-cell molecular map of mouse gastrulation and early organogenesis. *Nature* **566**, 490-495, doi:10.1038/s41586-019-0933-9 (2019).
- 21 Nowotschin, S. *et al.* The emergent landscape of the mouse gut endoderm at single-cell resolution. *Nature* **569**, 361-367, doi:10.1038/s41586-019-1127-1 (2019).
- 22 Butler, A., Hoffman, P., Smibert, P., Papalexi, E. & Satija, R. Integrating single-cell transcriptomic data across different conditions, technologies, and species. *Nat Biotechnol* **36**, 411-420, doi:10.1038/nbt.4096 (2018).
- 23 Lin, L. *et al.* The homeobox transcription factor MSX2 partially mediates the effects of bone morphogenetic protein 4 (BMP4) on somatic cell reprogramming. *J Biol Chem* **293**, 14905-14915, doi:10.1074/jbc.RA118.003913 (2018).

Reviewer #1 (Remarks to the Author):

The authors have satisfactorily responded to all my comments.

A final minor point: the authors should carefully check the text for typos, particularly throughout the methods section.

Reviewer #2 (Remarks to the Author):

In this revised manuscript by Ming and colleagues, the authors have provided answers to most of the raised comments.

Point-by-point response to the reviewers:

Reviewer #1:

Dear reviewer,

We sincerely appreciate your efforts for considering and reviewing our work again. We have checked and corrected them in the final version. We would like to express our gratitude for your insightful comments and suggestions which have significantly contributed to improve the quality of our work.

Reviewer #1:

Dear reviewer,

Thank you for your invaluable time and efforts dedicated to reviewing our manuscript. We are so happy that our previous revised manuscript has provided answers to most of your comments. We sincerely acknowledge your rigorous evaluation and comments, which have immensely improved our work.

Sincerely,

Duanqing